# The wide range of factors contributing to Wind Resource Assessment accuracy in complex terrain

Sarah Barber[1], Alain Schubiger[1], Sara Koller[2], Dominik Eggli[2], Alexander Radi[3], Andreas Rumpf[4], and Hermann Knaus[4]

[1]Eastern Switzerland University of Applied Sciences, Oberseestrasse 10, 8640 Rapperswil, Switzerland
[2]Meteotest AG, Fabrikstrasse 14, 3012 Bern, Switzerland
[3]Enercon GmbH, 14 E Rue des Clairières, 44840 Les Sorinières, France
[4]Hochschule Esslingen, Kanalstr. 33, 73728 Esslingen am Neckar, Germany

**Correspondence:** Sarah Barber (sarah.barber@ost.ch)

**Abstract.** Understanding the uncertainties of Wind Resource Assessments (WRA) is key to reducing project risks, and this is particularly challenging in mountainous terrain. In the academic literature, many 'complex flow' sites have been investigated, but they all focus on comparing wind speeds from selected wind directions, and do not focus on the overall Annual Energy Production (AEP). In this work, the importance of converting wind speed errors into AEP errors when evaluating wind energy projects is highlighted by comparing the results of seven different WRA workflows at five complex terrain sites. Although a systematic study involving the investigation of all possible varying parameters is not within the scope of this study, the results allow some of the different factors that could lead to this discrepancy to be identified. The wind speed errors are assessed by comparing simulation results to wind speed measurements at validation locations. This is then extended to AEP estimations (without wake effects), showing that wind profile prediction accuracy does not translate directly or linearly to AEP accuracy. This to the specific conditions at the site, to differences in workflow set-ups between the sites as well as to differences in workflow AEP calculation methods. The results demonstrate the complexity of the combined factors contributing to WRA errors - even without including wake effects and other losses. This means that the wind model that produces the most accurate wind predictions for a certain wind direction over a certain time period does not always result in the most suitable model for the AEP estimation of a given complex terrain site. In fact, the large number of steps within the WRA process often lead to the choice of wind model being less important for the overall WRA accuracy than would suggest by only looking at wind speeds. It is therefore concluded that it is vitally important for researchers to consider overall AEP - and all the steps towards calculating it - when evaluating simulation accuracies of flow over complex terrain. Future work will involve a systematic study of all the factors that could contribute to this effect.

# 1   Introduction

Understanding the uncertainties of Wind Resource Assessments (WRA) is key to reducing project risks. This is particularly challenging in mountainous terrain, e.g. Bowen and Mortensen (1996); Wood (1995); Pozo et al. (2017), which accounts for around 30% of the world's land surface (Sayre et al., 2018).

Several previous studies examine and compare the performance uncertainties of different wind modelling tools at mountainous, or 'complex' sites, including the Bolund Hill Blind Test (Bechmann et al., 2011; Berg et al., 2011), the Askervain Hill
Blind Test (Bao et al., 2018) and the Perdigão field test (Menke et al., 2019; Barber et al., 2020a). However, they are all limited to comparisons of wind speeds for chosen wind directions, or to time periods that are much shorter than those required for WRA.

In order to transform wind speeds into Annual Energy Production (AEP), which is required for WRAs, the expected wind speeds at a planned wind turbine location need to be described in the form of a frequency distribution. If the planned wind
turbine hub height differs from the height for which the wind speeds are available, a vertical extrapolation is required. The frequency distribution needs to be obtained for different wind direction sectors, because the wind flow at 'complex' sites varies with direction. As well as this, the frequency distributions need to be extrapolated for the expected lifetime of the wind turbine (usually 20 years). The resulting frequency distributions then need converting to power using the expected power curve of the planned wind turbine. The expected powers then need to be multiplied by time, added up and scaled for a one year time
period to estimate the AEP. If wake effects are expected, i.e. if the wind turbine is located within a wind farm, they need to be estimated using a wake model such as the Jensen model (Jensen, 1983), the Park model (Mortensen et al., 1998), the Larsen model(Larsen, 1988) and the dynamic wake meandering (DWM) model (Larsen et al., 2008). Other losses such as electrical losses, grid losses, curtailment losses as well as shutdowns due to maintenance and any environmental restrictions due to bats, birds, noise, flicker, etc. are also estimated. Due to the large number of separate steps, data types and organisations involved in
WRAs, it is challenging to accurately and robustly evaluate the accuracy of different tools or workflows for the entire process. A key obstacle is the lack of availability and suitability of relevant validation data.

The only previous studies examining the steps required for full WRAs are the set of 'CREYAP' exercises (Mortensen et al., 2015) and a US-based study on Annual Energy Prodution (AEP) errors (Lee and Fields, 2021). CREYAP stands for 'Comparison of Resource and Energy Yield Assessment Procedures' and is carried out by Ørsted and the Technical University
of Denmark. In the 2021 version, participants estimate the net energy yield for the Walney Extension wind farm, accounting for wind speed variations over time and across the site, and for turbine interaction losses[1]. A summary of the findings from the previous exercises concludes that most of the steps involved in WRA require significant improvement, and the study needs extending for complex terrain effects (Mortensen et al., 2015). The US-based study presents a very valuable literature review of the energy yield assessment errors across the global wind energy industry and provides a summary of how the wind energy
industry has been quantifying and reducing prediction errors, energy losses, and production uncertainties (Lee and Fields, 2021). In their work, a long-term trend indicating a reduction in the over-prediction bias was identified. Both of these studies

---

[1]https://windeurope.org/tech2021/creyap-2021/

provide valuable information; however, they are limited due to confidentiality issues connected with collecting data from the industry.

The goal of this work is therefore to examine and compare the accuracies of a range of simulations at different complex flow sites in terms of both wind speed and AEP. In order to achieve this, simulations with seven different wind modelling workflows at five different complex terrain sites were carried out, and the resulting wind speeds compared to measurements at a validation location. This was then extended to AEP estimations (without wake effects), and the differences examined. The workflows involved a range of wind modelling tools as well as a range of model set-ups and AEP calculation methodologies.

The paper is organised as follows: the methods applied in this work are described in Section 2, the results in terms of wind speed are presented in Section 3, the results are extended to AEP in Section 4, and the conclusions are discussed in Section 5.

## 2 Applied methods

### 2.1 The applied workflows

In this paper, the simulation set-ups are referred as 'workflows' to highlight the fact that each AEP estimation contains an entire workflow, i.e. many steps as well as the wind modelling part. For each workflow, the digital topography and surface roughness data for each site was downloaded from the satellite observation Copernicus database[2].

This work was part of a larger study developing a new decision process for selecting the WRA workflow that would expect to deliver the best compromise between skill and costs for a given wind energy project is developed, with a focus on complex terrain (Barber et al., 2022). For this study, a large range of different wind models and WRA set-ups were chosen in order to understand the relationship between skill and costs. This means that for the present paper, a systematic study investigating the effect of different factors on the AEP errors could not be carried out. This is the topic of future work.

#### 2.1.1 Underlying wind models

The workflows applied were associated with different wind models as follows:

- **WF-1: WindPro**: The industry-leading software suite for design and planning of wind farm projects[3]. It allows calibration data to be entered directly at the mast location and splits the flow into a total of 12 wind direction sectors. The turbulence intensity is calculated directly from the input met mast data standard deviations and mean values. In this work, the Linearized Flow Model was applied and no RIX corrections were applied.

- **WF-2: WindSim**: An industry software using a Computational Fluid Dynamics (CFD) model based on the PHOENICS code, a 3D Reynolds-Averaged Navier-Stokes (RANS) solver from the company CHAM. It allows calibration data to be entered directly at the mast location and splits the flow into pre-defined wind direction sectors. The turbulence intensity is calculated directly from the input met mast data standard deviations and mean values. In this work, the simulations

---

[2]https://land.copernicus.eu/
[3]www.wasp.dk

were conducted using the standard k-$\epsilon$ turbulence model (Rodi and Spalding, 1983). The boundary condition at the top was fixed pressure. A total of 12 wind direction sectors were used.

- **WF-3: ANSYS CFX**: An all-purpose CFD software. In this work, ANSYS CFX (version 19.2) was applied using an Unsteady RANS approach in combination with the standard k-$\epsilon$ model (Rodi and Spalding, 1983). The equation system was extended to an anelastic formulation whereby the density is influenced by buoyancy forces using the Boussinesq approximation (Montavon, 1998). The Earth's rotation was considered with additional terms in the momentum equation to describe the Coriolis force at a given angular velocity. Forested areas were considered through a canopy model (Liu et al., 1996). Individual roughness lengths of the canopy were attributed to each land use and are chosen according to the work of Wieringa (Wiernga, 1993). These methods have previously been applied successfully to other applications (Knaus et al., 2017). Static boundary conditions were applied around the resolved volume for the chosen wind direction and wind speed, forming a Taylor Spiral. Since the upper boundary condition of the domain is flat, but the ground surface is not, the mass flow rate was balanced at the top surface of the domain in order to prevent an acceleration due to mass conservation of the setup. Turbulence was introduced by entering a turbulence intensity of 10% as an input condition. In order to match the WindPro and WindSim simulations, a total of 12 wind direction sectors were simulated. This was implemented by developing a new simulation workflow using a Python script as described in Barber et al. (2020b).

- **WF-4: Fluent RANS**: ANSYS Fluent is a commercial CFD tool for modelling the flow in industrial applications and can be set up to solve the RANS equations, as well as for the Large Eddy Simulations (LES) or Detached Eddy Simulations (DES) approach. In this work, Fluent was first set up to solve the RANS equations with the SST k-$\omega$ turbulence model (Menter, 2012). The inlet wind speed profiles, turbulent kinetic energy and turbulent dissipation rate were imposed based on the roughness height according to Richards and Hoxey (1993). In order to match the WindPro and WindSim simulations, a total of 12 wind direction sectors were simulated. This was implemented by developing a new simulation workflow using a Python script as described in Barber et al. (2020b).

- **WF-5: Fluent SBES**: A technique within ANSYS Fluent called "Stress-Blended Eddy Simulation" (SBES) was applied. SBES is a new model that offers improved shielding of RANS boundary layers and a more rapid RANS-LES "transition", amongst other things. The RANS results were used to initialise SBES, which was performed unsteadily with an adaptive time stepping procedure ensuring CFL <= 1. After an additional unsteady initialisation, the wind speeds were averaged over 10 minutes. The wind rose measured at the met mast was used to choose the main wind directions, and only these directions were simulated. In order to introduce fluctuating velocities at the inlet, the Fluent Synthetic Turbulence Generator was used. In order to save computational power, all 12 wind direction sectors were not simulated. Instead, only the most frequently occurring sectors were simulated, and Fluent RANS results were used for the other sectors. In this work, two separate workflows were applied: for workflow WF-5a, three sectors were simulated with SBES and for workflow WF-5b, seven sectors were simulated with SBES. This was implemented by developing a new simulation workflow using a Python script as described in Barber et al. (2020b).

- **WF-6: PALM**: The model PALM is based on the non-hydrostatic, filtered, incompressible Navier-Stokes equations in Boussinesq-approximated form (an anelastic approximation is available as an option for simulating deep convection). Furthermore, an additional equation is solved for either the subgrid-scale turbulent kinetic energy using Large-Eddy Simulations (LES)[4]. In this work, due to the long computational time, not all sectors could be simulated. Instead, the most frequent sectors were simulated and complemented with the WindSim simulations. The TKE-1 Turbulence Model[4] was applied.

- **WF-7: E-Wind**: E-Wind solves the 3D RANS equations, with a modified k-$\epsilon$ turbulence closure. The governing equations are implemented in the open source toolbox OpenFOAM (Alletto et al., 2018). It is based on a steady-state two-equation RANS model and includes the effect of forests, Coriolis force, and buoyancy in the turbulence equations. It has only been applied to Site 3 in this work, due to the interest of the E-Wind developers, Enercon, in that site.

### 2.1.2 Simulation calibration

For each site, measurement data was available for at least two different locations, and the most available and reliable data at the location closest to the planned or existing wind turbine hub height(s) was chosen for calibration. For WindPro and WindSim, the calibration data could be directly input into the tool. For the CFD simulations, the calibration was undertaken for each wind direction sector simulated by inputting a generic logarithmic wind profile calculated using the roughness height value at the simulation input location (Richards and Hoxey, 1993). Then, the simulation results were scaled by a constant calibration factor, defined as the ratio between the simulated and the long-term corrected measured wind speed at the calibration location.

### 2.1.3 Simulation validation

For each site, the second measurement point was taken for the validation. The validation was carried out by first calculating the wind speed and direction at the validation location for each calibrated wind direction sector simulation, and then by comparing these values to the measurements.

### 2.1.4 Wind speed long-term extrapolation

For each site, the long-term corrected measured wind speed was obtained at both the calibration and validation locations by firstly averaging the wind speeds over a time period for which valid data was available for both the calibration and validation measurements. Secondly, these average wind speeds were extrapolated for a 20-year period using the Measure-Correlate-Predict method with long-term reference data. This data was either obtained from a nearby met mast or from MERRA-2 data[5], as summarised in Table 1.

---

[4]https://palm.muk.uni-hannover.de/trac/wiki/palm
[5]https://gmao.gsfc.nasa.gov/reanalysis/MERRA-2/

**Table 1.** Summary of the long-term extrapolation methods for each site

| Site | Site 1 | Site 2 | Site 3 | Site 4 | Site 5 |
|---|---|---|---|---|---|
| **Reference data** | MERRA (lon 10, lat 48.5; 10hPa above ground) | Met mast STIG1 50m | MERRA (lon -3.75, lat 43; 10hPa above ground) | None | Unnamed weather station |
| **Reference data time period** | 01.01.2010 - 31.12.2019 | 01.06.2007 - 31.05.2016 | 01.01.2010 - 30.09.2020 | None | 1998-2018 |
| **Measured data** | Met mast at site (50 m) | STIG2 met mast (83 m) | Met mast WEST (50 m) | None | Unnamed met mast at site |
| **Measured data time period** | Mar. 2015 - Feb. 2016 | Dec. 2013 - Jan. 2016 | Sep. 2009 - Sep. 2010 | None | 6 months |
| **MCP method** | Linear least-square | Linear least-square | Linear least-square | None | Linear least-square |
| **Correlation coefficient $R^2$** | 0.96 | 0.94 | 0.96 | None | 0.71 |

### 2.1.5 AEP calculation

The wind industry tools applied in this work, WindPro and WindSim, as well as E-Wind, offer a user interface containing all the steps required to get from wind measurements to AEP predictions. However, the other CFD tools, Fluent, CFX and PALM needed to be extended in order to calculate the AEP. In previous work (Barber et al., 2020b), automated workflows for carrying out wind flow modelling in complex terrain were developed for the commercial CFD tools ANSYS Fluent and ANSYS CFX. These processes included a pre-processing tool, a meshing script, a simulation script and a pre-processing tool. It was shown that the total simulation costs, including model set-up and post-processing effort costs, could be reduced by a factor of 12 for Fluent and seven for CFX through this automation. For PALM, a slightly different method was applied in order to reduce the computational time in a pragmatic way. This involved firstly searching for a time period in the met mast data for which the wind speed remained relatively stable at the average wind speed for each wind direction sector. For this time period, COSMO-D2 data[6] was taken for the PALM boundary conditions by reading the data from a dynamic input file (offline nesting). Finally, this time period is simulated with PALM and the average wind speed is extracted for the calibration and validation positions at the reference height.

In the present work, these workflows were extended to include AEP estimations. The AEP was calculated at the validation location(s) as follows:

---

[6]http://www.cosmo-model.org/content/tasks/operational/dwd/default$_d$2.htm

- Speed-up factors between the validation location(s) and the calibration location were calculated from the simulation results for each wind direction sector (in this case, 12 sectors). The speed-up factor is defined as the ratio between the wind speed at the validation location to the wind speed at the calibration location.

- Option 'Time-series method':

  - The amount of wind turning between the validation location(s) and the calibration location was calculated from the simulation results for each wind direction sector.

  - Based on the wind direction sector, each measured ten-minute averaged wind speed at the calibration location was multiplied by the relevant speed-up factor to obtain the expected wind speed at the validation location for that ten minute period.

  - For sectors with a wind direction turning larger than +/- 15°, each data point was moved to the relevant new sector.

  - The linearly-interpolated bin-averaged power curve provided by the manufacturer with $1\,\mathrm{ms}^{-1}$ wide bins was used to obtain the expected power production for every ten-minute wind speed at the validation location(s). For this, it was assumed that the wind speed at the validation location remained constant over the entire wind turbine rotor area. The Rotor Equivalent Wind Speed was not used because the commercial tools do not use this method and a fair comparison was required.

  - These powers were each multiplied by 10 minutes and added up to obtain the total energy production over the measurement period.

- Option 'Weibull method':

  - A frequency distribution of the calibration location data for was created for each sector.

  - A Weibull distribution was fitted to each frequency distribution.

  - The Weibull distributions were scaled for each sector using the simulated speed-up factors. The shape factor was kept constant.

  - The scaled Weibull distributions were multiplied by the bin-averaged power curve provided by the manufacturer with $1\,\mathrm{ms}^{-1}$ wide bins to obtain the total energy production over the measurement period.

- The total energy production was converted to AEP by scaling the energy production in each wind direction sector linearly with the total measurement time.

The three research partners in this project, OST (Eastern Switzerland University of Applied Sciences), HSE (Esslingen University of Applied Sciences) and Meteotest, created their own separate workflows for this AEP calculation. For validation purposes, the workflows were firstly compared to each by entering the same simulation results from the WindSim model at one site (Site 1, see description in Section 2.2). An overview of the results is shown in Table 2. As well as comparing the results

between organisations, the 'Time-series method' and the 'Weibull method' are compared. Finally, the effect of interpolating the power curve or simply using the $1\,\mathrm{ms}^{-1}$ wide bins is compared.

As can be seen in the right-hand column, which shows the percentage difference in AEP for each calculation compared to calculation number 1, the variation in AEP between the different partners is less than 5%, giving us a general confidence in the methods. The largest variation is between the 'Weibull method' and the 'Time series method' (3.4% for the OST method). The difference between the power curve bins and the interpolation is small (1.2% for the OST method, 0.6% for the HSE method, 0.1% for the WindSim method). It should be noted that 5% is not an insignificant amount of energy, and the variations between methods are surprising to the authors. A significant effort in comparing the methods and correcting small errors did not lead to a reduction in these differences.

**Table 2.** Summary of AEP comparisons for the different methods used in this paper

| Calculation number | Organisation | Method | Power curve | AEP (kWh) | % difference in AEP from calculation 1 |
|---|---|---|---|---|---|
| 1 | OST | Weibull | Bins | 1,124,125 | - |
| 2 | OST | Time series | Bins | 1,073,634 | -4.5% |
| 3 | HSE | Time series | Bins | 1,059,076 | -5.8% |
| 3.1 | HSE | Time series | Interpolated | 1,067,386 | -5.1% |
| 4 | WindSim | Time series | Bins | 1,075,354 | -4.3% |
| 4.1 | WindSim | Time series | Interpolated | 1,075,830 | -4.3% |
| 5 | WindSim | Weibull | Bins | 1,107,025 | -1.5% |
| 5.1 | WindSim | Weibull | Interpolated | 1,065,270 | -5.2% |
| 6 | Meteotest | Time series | Bins | 1,179,359 | 4.9% |
| 6.1 | Meteotest | Time series | Interpolated | 1,174,415 | 4.5% |

### 2.1.6 Summary of workflows

The different set-ups for each workflow are summarised in Table 3. As well as describing the underlying wind tool, the flow model, the turbulence model, the number of wind directions simulated, the turbulence intensity (TI) input method and the calibration method applied, the last three rows compare the common application range of the tools as well as their relative simulation time and relative set-up complexity, in order to give the reader a feel for the type of workflows applied. The relative simulation time and set-up complexity are key factors when choosing the most optimal workflow for a given application. This topic has been studied in detail in a separate paper by the same authors (Barber et al., 2022).

**Table 3.** Summary of the seven workflows used in this study

| Workflow | WF-1 | WF-2 | WF-3 | WF-4 | WF-5 | WF-6 | WF-7 |
|---|---|---|---|---|---|---|---|
| **Underlying wind tool** | WindPro | WindSim | ANSYS CFX | ANSYS Fluent | ANSYS Fluent | PALM | OpenFOAM |
| **Flow model** | Linearized Flow Model | RANS CFD | URANS CFD | RANS CFD | SBES CFD | LES CFD | RANS CFD |
| **Turbulence model** | - | k-$\epsilon$ | k-$\epsilon$ | SST k-$\omega$ | - | - | k-$\epsilon$ |
| **Number of sectors** | 12 | 12 | 12 | 12 | 12 | 12 | 12 |
| **TI input** | Calibration data[7] | Calibration data | Input TI = 10% | TKE imposed [8] | TKE imposed | Calibration data | Input TI = 10% |
| **Calibration method** | Direct[9] | Direct | One-point[10] | One-point | One-point | One-point | One-point |
| **Common application range** | Only for flat terrain[11] | Non-flat terrain | Non-flat terrain | Non-flat terrain | Non-flat terrain | Complex weather | Non-flat terrain |
| **Relative simulation time** | Low | Medium | Medium | Medium | High | Very high | Medium |
| **Relative set-up complexity** | Low | Low | Medium | Medium | Medium | Very high | Low |

---

[7]Calculated directly from calibration data using mean and standard deviation of wind speed

[8]Turbulent kinetic energy and turbulent dissipation rate imposed based on roughness height at input location from roughness map

[9]Calibration data entered directly

[10]One-point calibration of generic log profile using roughness height

[11]Usually defined as slopes below 30% Bowen and Mortensen (1996)

## 2.2 The simulated sites

An overview of the simulated sites is shown in Fig. 1 and in Table 9, except for Site 5, which is confidential. The details and the workflow set-ups for each site are described in the individual sections below.

### 2.2.1 Site 1

Site 1 is a complex terrain, partly-forested site close to Stoetten in southern Germany, whose central feature is a steep incline above 30% and a main wind direction of W. Wind speed and direction data was available from a met mast located about 1 km away from the incline, as well as a lidar, as described in Schulz et al. (2014). In this paper, the met mast data from four cup anemometers with an accuracy of 1% and three 3D sonic anemometers with an accuracy of 1.5% were used for calibration, and the wind speed data recorded using the SWE[12]-Scanner, a fast pulsed lidar wind scanner based on a Leosphere Windcube V1 system with an adapted scanner unit, were used for validation. The coordinates of the calibration location are (4309721.715, 2838928.447) (coordinate system EPSG3035). For the measurement campaign used in this work, the lidar was positioned approximately 300 m west of the met mast for approximately one year (March 2015 - February 2016). An overview of the site and the wind rose from the met mast at 98 m over the entire measurement period are shown in Fig. 1. S&G Engineering SG750.54 wind turbines with a hub height of 100 m are planned at the site. The details of the applied workflows are shown in Table 4. WF-7 was not applied to this site.

**Table 4.** Set-up of the workflows at Site 1

| Criteria | WF-1 | WF-2 | WF-3 | WF-4 | WF-5 | WF-6 |
|---|---|---|---|---|---|---|
| **Grid dimensions** | 9x9 km | 9x9x5 km | 20x22x2.5 km | 10x10x1.5 km | 10x10x1.5 km | 6x6x6.5 km |
| **Horizontal resolution** | 25 m | 25 m | 25 m | 20 m | 20 m | 10 m |
| **Number of cells** | 129,600 | 4.5mil | 17mil | 20mil | 20mil | 54mil |

### 2.2.2 Site 2

Site 2 is an existing wind farm site in Norway surrounded by a terrain of hills, lakes and forests. The wind farm consists of seven Siemens SWT-DD-130 wind turbines with a hub height of 115 m located on a small hill as shown on Fig. 1. The circles on this figure refer to met masts and wind turbines. Another wind farm consisting of 22 wind turbines is located on the hill to the north of the wind farm. The two main wind directions are SE and NW, as can be seen on the wind rose from the met mast STIG2 in Fig. 1. In this project, the two met masts close to the wind turbines, 'STIG1' and 'STIG2', are used. For STIG1, wind speed measurement data is available from Thies First Class Cup Anemometers at five heights up to 50 m from June 2007 to May 2016 (9 years). For STIG2, wind speed measurement data from cup anemometers is available at five heights up to 83 m from December 2013 to January 2016 (2 years). STIG1 has three sensors at very similar heights (48 m, 49 m and 50 m),

---

[12]Stuttgarter Lehrstuhl fuer Windenergie

whereas STIG2 has two sensors at similar heights (82 m and 83 m). An examination of the time series of the measurement data showed that the sensors at 47 m, 48 m and 82 m contained some errors, and therefore the sensors at these heights were not used. STIG2 at 83 m is taken as the calibration mast, with coordinates of (4086674.443, 3958009.528) (coordinate system EPSG3035). The details of the applied workflows are shown in Table 5. WF-3 and WF-7 were not applied to this site.

**Table 5.** Set-up of the workflows at Site 2

| Criteria | WF-1 | WF-2 | WF-4 | WF-5 | WF-6 |
|---|---|---|---|---|---|
| Grid dimensions | 25x25 km | 5x5x5 km | 10x10x1.5 km | 10x10x1.5 km | 20x10x4 km |
| Horizontal resolution | 25 m | 25 m | 20 m | 20 m | 20 m |
| Number of cells | 1.0mil | 1.9mil | 20mil | 20mil | 1.6mil |

### 2.2.3 Site 3

Site 3 is an existing wind farm site in Spain situated in complex terrain. The wind farm consists of 15 Enercon E-40 wind turbines with a hub height of 50 m located at the top of a steep ridge as shown in Fig. 1. The circles on this figure refer to met masts and wind turbines. The two main wind directions are SW and SWW, as can be seen on the wind rose in Fig. 1. In this project, data from two met masts 'EAST' and 'WEST' were used. For both of these masts, measurement data was provided from Thies First Class Cup Anemometers at a range of heights up to 50 m from September 2009 to September 2010. Due to the high amount of overlapping data points (48,941), they can be used for the calibration and validation data reliably. The WEST mast was used for calibration and the EAST mast for validation. The coordinates of the calibration location are (442929.200, 4741536.600) (coordinate system EPSG3035). The details of the applied workflows are shown in Table 6. All seven workflows were applied to this site.

**Table 6.** Set-up of the workflows at Site 3

| Criteria | WF-1 | WF-2 | WF-3 | WF-4 | WF-5 | WF-6 | WF-7 |
|---|---|---|---|---|---|---|---|
| Grid dim. | 20x20 km | 10x10x8 km | 20x20x4 km | 10x10x1.5 km | 10x10x1.5 km | 18x18x10 km | 20 km dia. |
| Hor. res. | 25 m | 25 m | 25 m | 20 m | 20 m | 25 m | 50 m |
| No. cells | 640,000 | 13.8mil | 1.6mil | 20mil | 20mil | 17.6mil | 600,000 |

### 2.2.4 Site 4

Site 4 is the existing St. Brais wind farm site in Switzerland situated in complex terrain. The wind farm consists of two Enercon E-82 wind turbines with a hub height of 78 m located as shown in Fig. 1. The main wind direction is SW, as can be seen on the wind rose on the figure. In this project, wind speed and direction data from the two wind turbines 'WEA1' and 'WEA2' were

used, because no met mast data was available. The measurement location on the nacelle of the wind turbines (hub height 78 m) means that the data will be influenced by the wind turbine rotors, and the measurements will not correspond to the freestream wind speed. It is not known if the measurement data include any kind of correction for this effect or not. This means that the data needs to be used carefully in this project, and the results will only be able to be used indicatively. This is discussed further in Section 4. The absolute values of the wind speeds may not be correct, but the ratios will still be valid, assuming that there are no large discrepancies in the operation of the wind turbines. For both of these wind turbines, measurement data was provided from January 2010 to December 2020, a time period of 10 years. Due to the high amount of overlapping data points (554,045), they can be used for the calibration and validation data reliably. A long-term extrapolation was not done because the measurement period is already very long. The wind turbine WEA1 was used for calibration and WEA2 for validation. The coordinates of the calibration location are (574452.000, 239147.000) (coordinate system EPSG21781). A good correlation between the WEA1 and WEA2 data could be seen. The details of the applied workflows are shown in Table 7. The workflows WF-3, WF-6 and WF-7 were not applied.

**Table 7.** Set-up of the workflows at Site 4

| Criteria | WF-1 | WF-2 | WF-4 | WF-5 |
|---|---|---|---|---|
| Grid dimensions | 9x9 km | 10x10x10 km | 10x10x1.5 km | 10x10x1.5 km |
| Horizontal resolution | 25 m | 25 m | 20 m | 20 m |
| Number of cells | 129,600 | 10.2mil | 20mil | 20mil |

### 2.2.5    Site 5

Site 5 is a planned site that cannot be described in detail here for confidentiality reasons. However, the results are still included in this work in the form of comparisons. The site is located in hilly terrain. Data from two met masts over a time period of six months is available. The met mast closest to the planned wind turbines was chosen for the calibration point. The details of the applied workflows are shown in Table 8. The workflows WF-3, WF-6 and WF-7 were not applied.

**Table 8.** Set-up of the workflows at Site 5

| Criteria | WF-1 | WF-2 | WF-4 | WF-5 |
|---|---|---|---|---|
| Grid dimensions | 9x9 km | 10x10x10 km | 10x10x1.5 km | 10x10x1.5 km |
| Horizontal resolution | 25 m | 25 m | 20 m | 20 m |
| Number of cells | 129,600 | 10.2mil | 20mil | 20mil |

**Table 9.** Details of each site simulated in this work

| Site | Site 1 | Site 2 | Site 3 | Site 4 | Site 5 |
|---|---|---|---|---|---|
| **Location** | Germany | Norway | Spain | Switzerland | - |
| **Main wind direction** | W | SE and NW | SW and SWW | SW | - |
| **Calibration location** | (4309721.715, 2838928.447) | (4086674.443, 3958009.528) | (442929.200, 4741536.600) | (574452.000, 239147.000) | - |
| **Coordinate system** | EPSG3035 | EPSG3035 | EPSG3035 | EPSG21781 | - |
| **Calibration** | Met mast (93 m) | STIG2 met mast (83 m) | Met mast WEST (50 m) | Wind turbine WEA1 | Unnamed met mast at site |
| **Validation** | Fast pulsed lidar wind scanner (100 m) | STIG1 met mast (83 m) | Met mast EAST (50 m) | Wind turbine WEA2 | Unnamed met mast at site |
| **Measurement time** | Mar. 2015 to Feb. 2016 | Dec. 2013 to Jan. 2016 | Sep. 2009 to Sep. 2010 | Jan. 2010 to Dec. 2020 | 6 months |
| **Wind turbine type** | S&G Engineering SG750.54 | Siemens SWT-DD-130 | Enercon E-40 | Enercon E-82 | - |
| **Wind turbine hub height** | 100 m | 115 m | 50 m | 78 m | - |

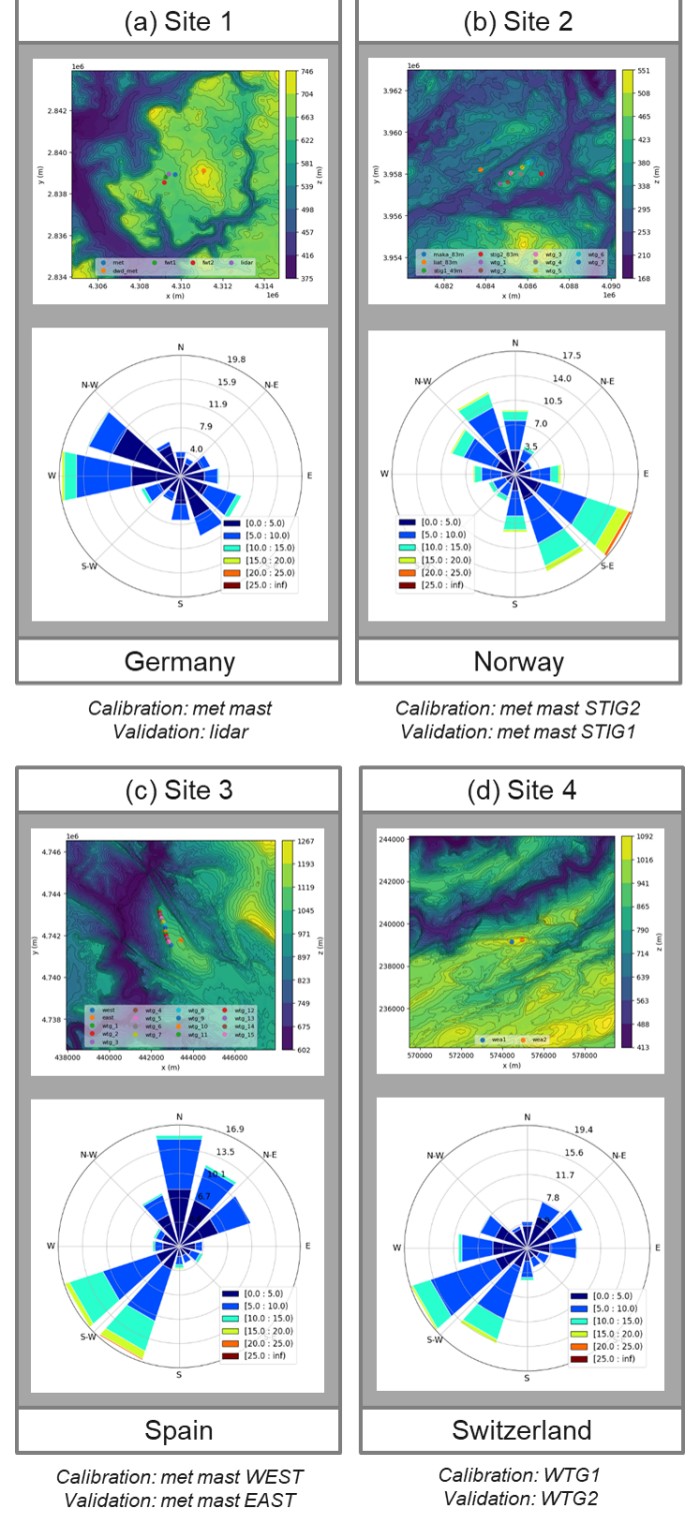

**Figure 1.** An overview of four of the five sites simulated in this work (Site 5 is confidential): (a) Site 1, (b) Site 2, (c) Site 3 and (d) Site 4

## 3 Simulation results - wind speeds

For each site, a total of 12 simulations were carried out - one for each 30° wind direction sector. In this section, the simulation results are shown in terms of absolute differences between the measurements and simulations at the calibration and validation locations for each site. For locations with data at more than one height, the Root Mean Square Error of all measured heights compared to simulations is used as a comparison.

Detailed simulation results for each site can be found in the final project report (Barber et al., 2021). This includes speed-up factors and turning between calibration and validation locations, comparisons of simulated and measured wind speed and direction profiles for all wind directions and wind speed contours.

### 3.1 Site 1

The average RMSE values between the simulations and the measurements over all 12 wind direction sectors are shown in Fig. 2. The top row ((a) and (b)) shows the calibration location and the bottom row ((c) and (d)) the validation location. The left-hand plots ((a) and (c)) show the results of a simple averaging of the RMSE values over the 12 wind direction sectors, whereas the right-hand plots ((b) and (d)) show RMSE averages weighted for the wind speed frequency distribution measured at the calibration location. This weighted averaging gives more weighting to more frequent wind speeds in an attempt to provide a number more relevant for the AEP. For this site, it can be seen that the weighted averaging does not have a large effect on the results. This indicates that the errors of the most frequent wind directions are close to the average value.

The different bars in each plot represent different measurement heights that have taken to calculate the RMSE, with the number of points increasing with increasing darkness of the colours. The light green colour refers to just one point and therefore the RMSE is equal to the absolute difference between measured and simulated wind speed. This number is very low for the calibration location as expected because of the calibration method described in Section 2.1,2. The measurement heights used for these calculations were 10 m, 25 m, 50 m, 75 m and 100 m. However, the error is not exactly equal to zero because the grid point in the simulation mesh nearest to the calibration position was taken for the calibration. This had a slight deviation in both the horizontal and vertical direction.

For the calibration location, the variable 'RMSE 25-100m' refers to the RMSE value using all the measurement heights except for the lowest point, because this point is difficult to simulate accurately and also less relevant for the AEP calculation as it is not inside the rotor area of the planned wind turbines. 'RMSE 10-100m' refers to the RMSE value using all the measurement heights. The values increase with an increasing number of points, because the simulations become less accurate close to the ground. This is expected, especially for heights below 25 m. In general, the results at the calibration location indicate that the wind speed profile has not been captured well for all workflows, even if the lower points are removed.

For the validation location, the variable 'RMSE 75-125m' refers to the RMSE value using only the heights within the rotor area of the planned wind turbines, and 'RMSE 50-150m' refers to the RMSE value using all the measurement heights. The values increase with an increasing number of points, because the simulations become less accurate close to the ground. This is expected, especially for heights below 25 m. However, this effect is not as large as for the calibration location. In general,

the entire profile is predicted fairly inaccurately, which is surprising due to the close location of the calibration and validation location (see Fig. 1). Another thing to notice about the RMSE values at the validation location is the particular low accuracy of WF-1 (WAsP) compared to the other workflows.

Examination of the absolute wind speed contours from the Fluent RANS simulations, shown in Fig. 7, indicate that the wind speeds are not expected to deviate significantly between the calibration and validation locations at any of the three main wind directions. The reasons for these validation errors of almost $1\,\mathrm{ms^{-1}}$ are not entirely clear and could be due to a measurement error. This is being further investigated.

### 3.2  Site 2

For Site 2, measurement data was only available at one height. Therefore the average absolute differences between the measurements and simulations over all 12 wind direction sectors at (a) the calibration location and (b) the validation location at hub height are shown in Fig. 3. The different bar colours represent simple and weighted averaging of the wind direction sectors, as introduced in the previous section. The error at the calibration location is very small, remaining below $0.006\,\mathrm{ms^{-1}}$ for each workflow. For the validation location, the errors all remain below $1\,\mathrm{ms^{-1}}$. WF-1 (WindPro) and WF-2 (WindSim) are closer to the measurements than WF-5 (Fluent) and WF-6 (PALM). For these three workflows, the order of the validation errors are similar to Site 1. Examination of the absolute wind speed contours at a height of 100 m above the ground for the three most common wind directions calculated using Fluent RANS in Fig. 8 indicates that varying validation errors between workflows could be caused by varying capabilities of capturing the flow over the hill, especially in the 30° direction. This is discussed in more detail in the final project report (Barber et al., 2021). The weighted averaging affects both WF-1 (WindPro) and WF-2 (WindSim), but for WF-1 it increases the error and for WF-2 it decreases the error. The reason for this is not clear and is investigated in Section 4.

### 3.3  Site 3

For Site 3, measurement data was only available at one height. Therefore the average absolute differences between the measurements and simulations over all 12 wind direction sectors at (a) the calibration location and (b) the validation location are shown in Fig. 4. For the calibration location, the errors for all workflows are all below $0.08\,\mathrm{ms^{-1}}$. These errors are slightly larger than the other sites because the discrepancy between the actual calibration position and the nearest grid point is larger. For the validation location, the errors all remain between about $0.4\,\mathrm{ms^{-1}}$ and $0.7\,\mathrm{ms^{-1}}$, a similar range to Site 2. However, the difference between workflows is less significant. WF-4 (Fluent RANS) and WF-5 (Fluent SBES) are generally more accurate. The weighted averaging generally has a small influence on the results. For these three workflows, the order of the validation errors are similar to Site 1. Examination of the absolute wind speed contours at a height of 100 m above the ground for the three most common wind directions calculated using Fluent RANS in Fig. 9 indicates that varying validation errors between workflows could be caused by varying capabilities of capturing the flow over the hill, especially in the 30° direction. This is discussed in more detail in the final project report (Barber et al., 2021).

### 3.4 Site 4

For Site 4, measurement data was only available at one height. Therefore the average absolute differences between the measurements and simulations over all 12 wind direction sectors at (a) the calibration location and (b) the validation location are shown in Fig. 5. For the calibration location, the errors for all workflows are all below $0.06\,\mathrm{ms}^{-1}$, for the same reasons as described above. For the validation location, the errors are even lower than the previous sites, remaining below $0.5\,\mathrm{ms}^{-1}$. Similarly to Site 2, WF-1 (WindPro) and WF-2 (WindSim) are closer to the measurements than WF-4 (Fluent RANS) and WF-5 (Fluent SBES). Again, the weighted averaging generally has a small influence on the results. The absolute wind speed contours at a height of 100 m above the ground for the three most common wind directions calculated using Fluent RANS are shown in Fig. 10. Similarly to Site 1, the validation error is unexpectedly large considering the small distance and difference in height between the calibration and validations locations, and perhaps reflects the poor quality of measurement data from the wind turbines compared to the met mast measurements at the other sites.

### 3.5 Site 5

For Site 5, measurement data was only available at one height. Therefore the average absolute differences between the measurements and simulations over all 12 wind direction sectors at (a) the calibration location and (b) the validation location are shown in Fig. 6. For the calibration location, the errors for all workflows are all below $0.02\,\mathrm{ms}^{-1}$, for the same reasons as described above. For the validation location, the errors are similar to Sites 1-3, remaining between $0.5\,\mathrm{ms}^{-1}$ and $1\,\mathrm{ms}^{-1}$. The significant distance between the calibration and validation locations on two separate hills and the corresponding differences in flow modelling capabilities over these hills could cause these validation errors. The wind speed contours cannot be shown here due to confidentiality reasons. Similarly to Site 3, the difference between the workflows is not very large. However, the effect of the weighted averaging is not insignificant. This was found to be because the largest deviations from the measurements occurred at the more frequent wind speeds, and is discussed further in Barber et al. (2021).

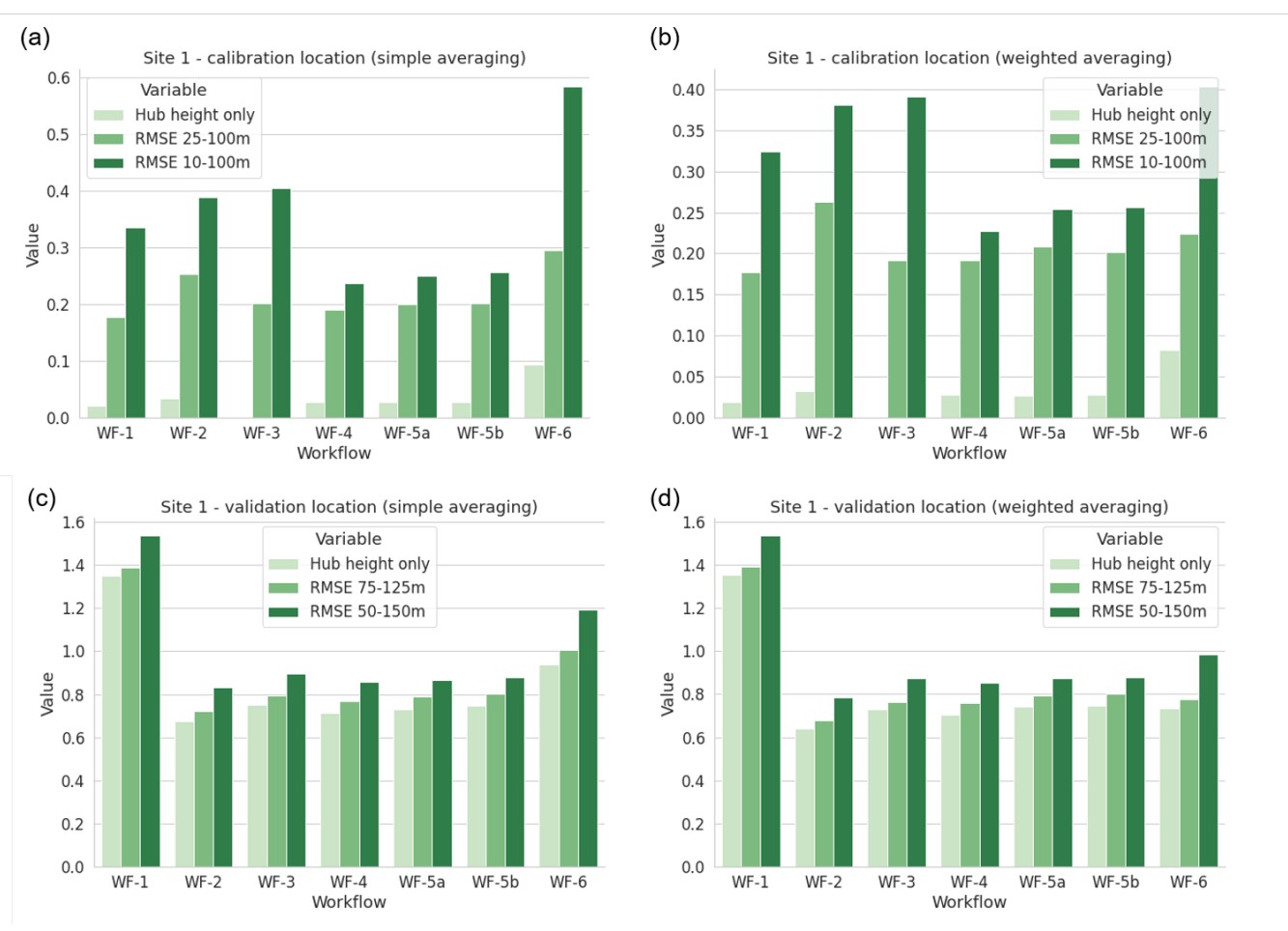

**Figure 2.** Site 1 - RMSE values between the measurements and simulations at (a) the calibration location and (b) the validation location for different numbers of measurement heights, for simple and weighted wind direction sector averaging $(\mathrm{ms}^{-1})$

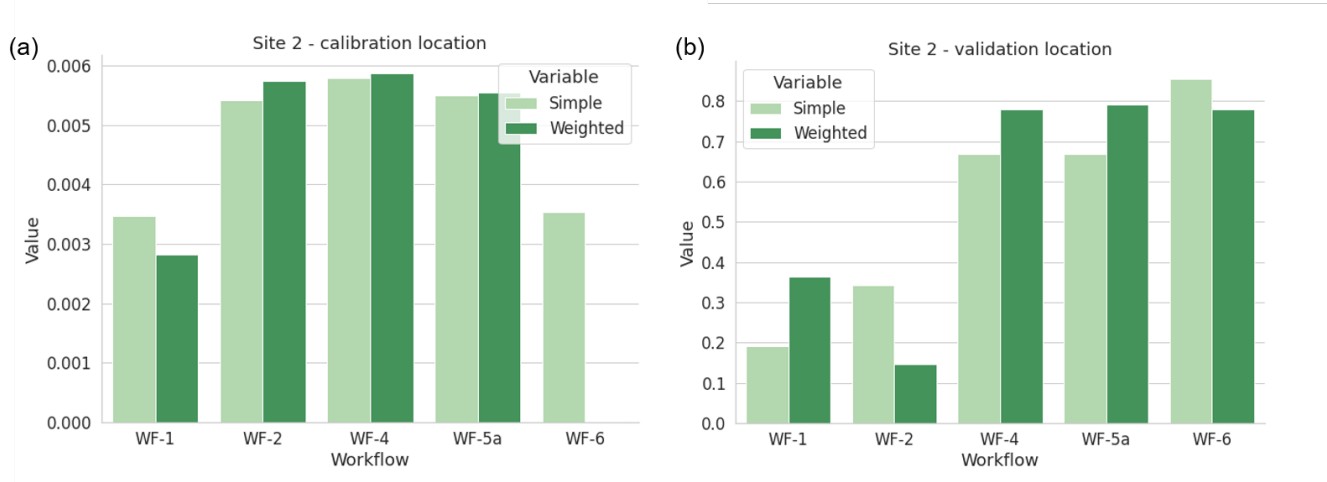

**Figure 3.** Site 2 - absolute differences between the measurements and simulations at (a) the calibration location and (b) the validation location ($\mathrm{ms^{-1}}$)

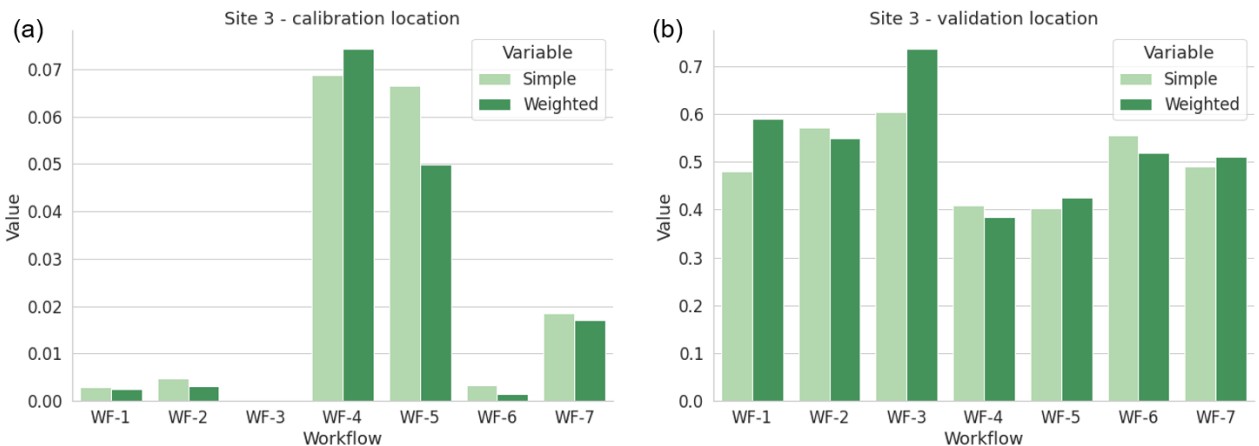

**Figure 4.** Site 3 - absolute differences between the measurements and simulations (a) the calibration location and (b) the validation location ($\mathrm{ms^{-1}}$)

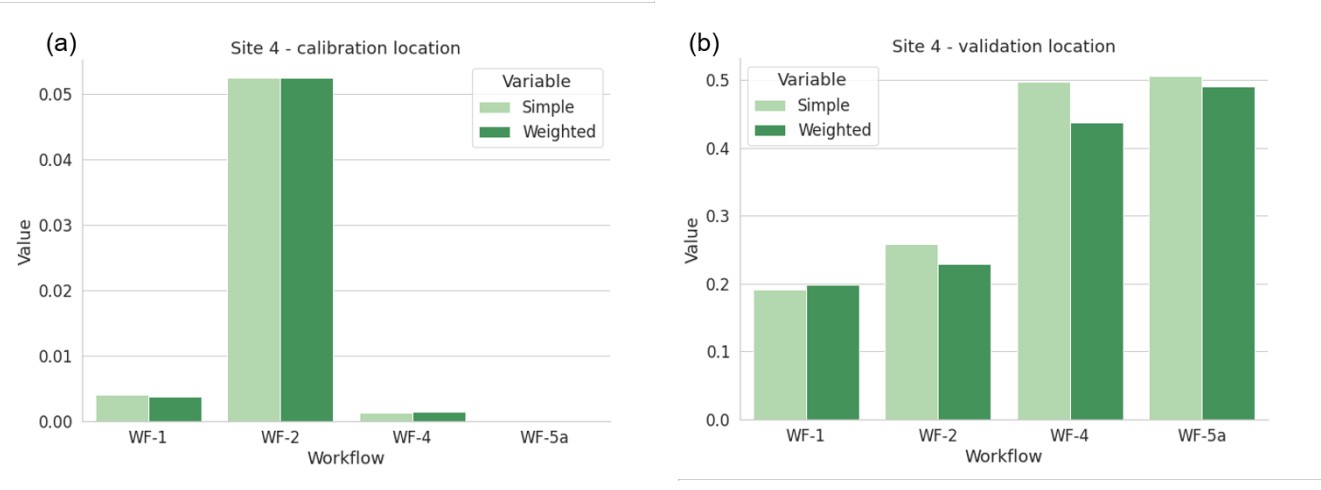

**Figure 5.** Site 4 - absolute differences between the measurements and simulations at (a) the calibration location and (b) the validation location $(ms^{-1})$

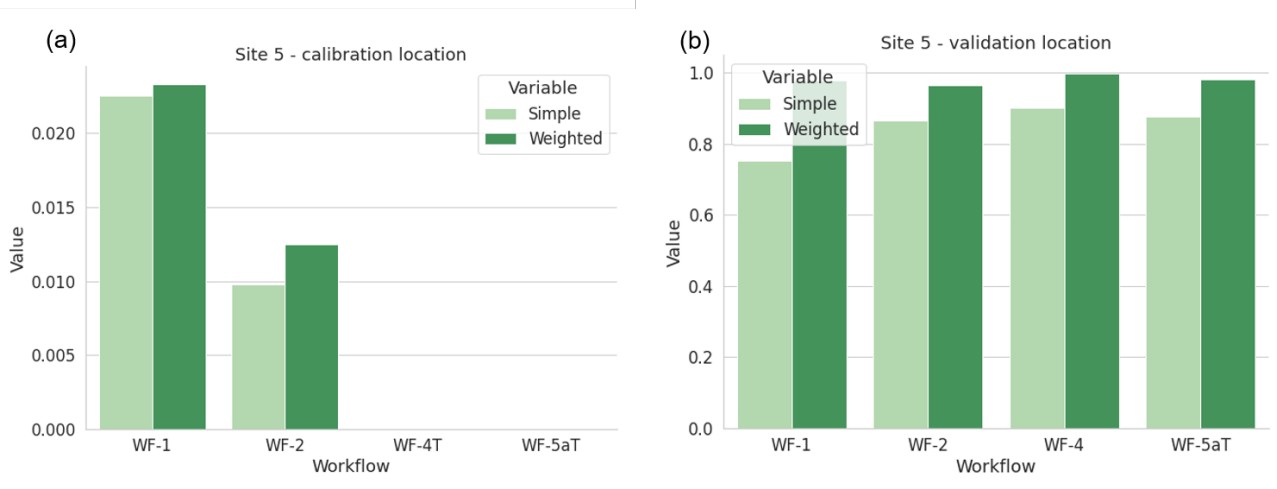

**Figure 6.** Site 5 - absolute differences between the measurements and simulations at (a) the calibration location and (b) the validation location $(\text{ms}^{-1})$

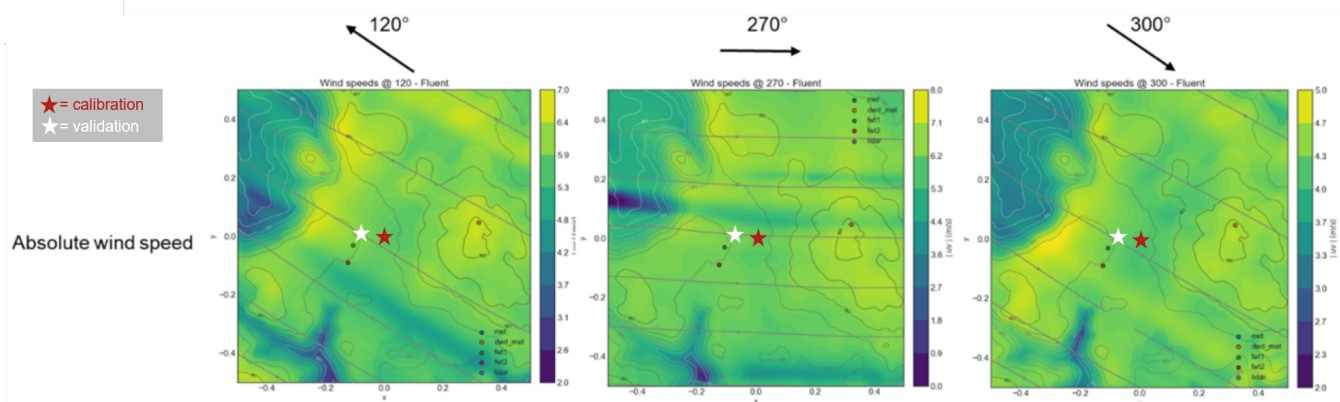

**Figure 7.** Site 1 - absolute velocity contours for the three main wind directions simulated in Fluent RANS at 100 m above ground level

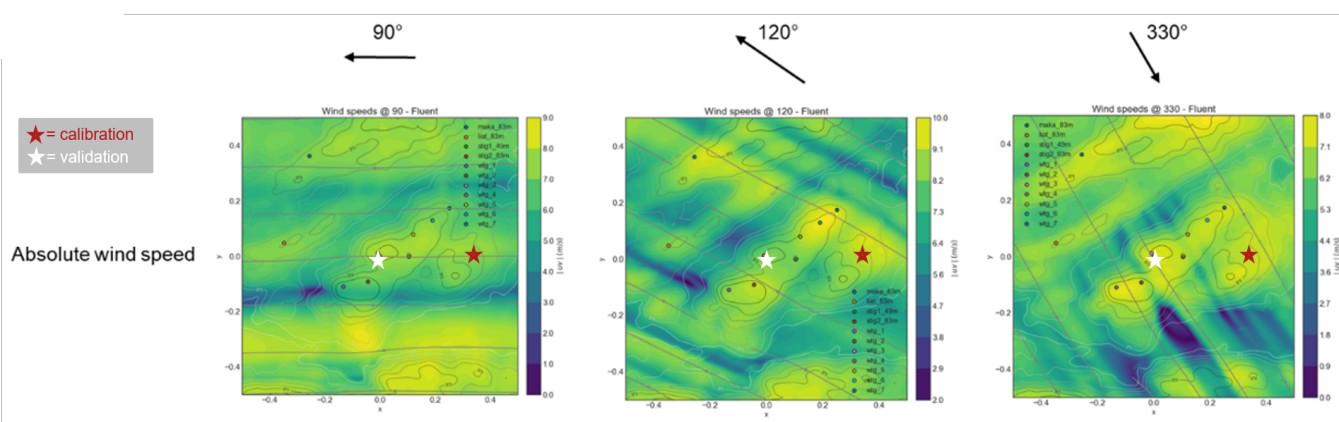

**Figure 8.** Site 2 - absolute velocity contours for the three main wind directions simulated in Fluent RANS at 100 m above ground level

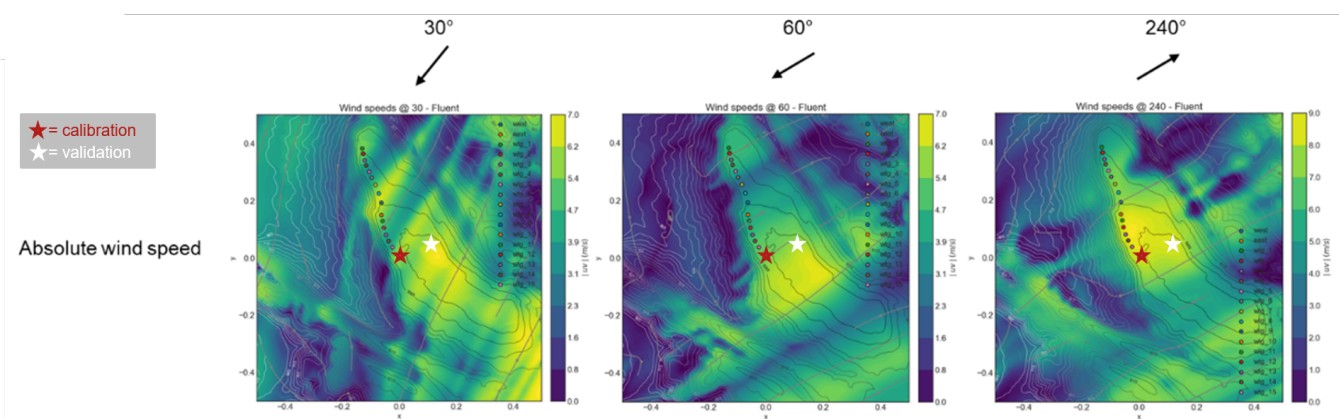

**Figure 9.** Site 3 - absolute velocity contours for the three main wind directions simulated in Fluent RANS at 100 m above ground level

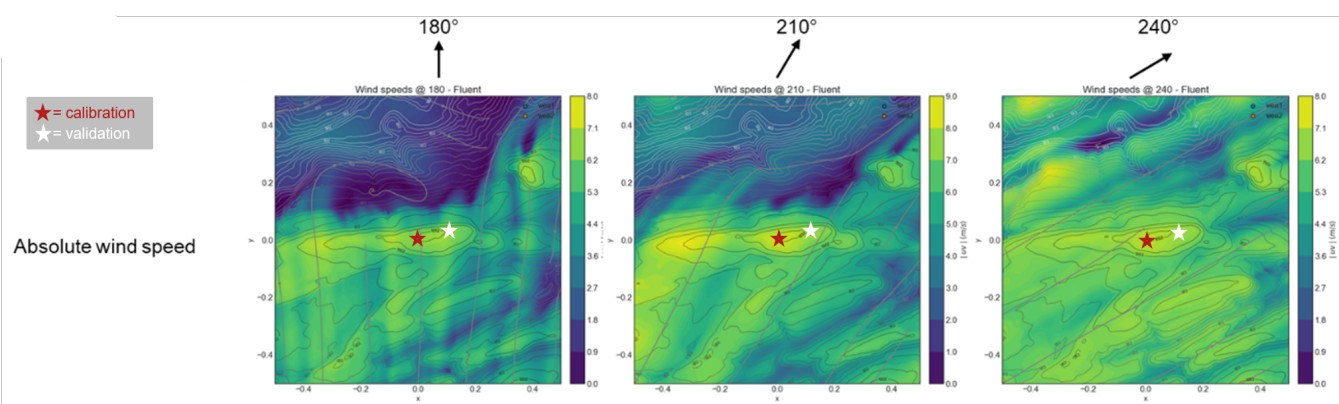

**Figure 10.** Site 4 - absolute velocity contours for the three main wind directions simulated in Fluent RANS at 100 m above ground level

# 4 Simulation results - Annual Energy Production (AEP)

In this section, the results at the validation location for each site are shown in terms of AEP, which was calculated from the simulation results for each workflow as described in Section 2. There are no AEP measurements to compare the results with, so instead each result was compared to a 'theoretical' value of AEP. This was determined using the AEP 'Time-series method' described in Section 2.1.4, but instead of using the simulation results to scale the calibration data time series, the measured time series at the validation location was used directly. This allowed the effect of the simulated differences in wind speeds to be transferred to differences in AEP. Throughout this work, AEP refers to the gross production, i.e. without any losses or wake effects. Any workflow name with the letter "T" added to it indicated that flow turning effects were considered, as described in Section 2.1.5, i.e. WF-3 refers to workflow WF-3 without turning considered, and WF-3T with turning. in the figure refers to the calculation with the consideration of turning effects.

## 4.1 Site 1

The wind turbine chosen for the AEP calculation at the validation location of Site 1 was a S&G Engineering SG750.54 with a hub height of 100 m, because this is the type planned for this site (construction planned in 2022). This wind turbine has a rated power of 750 kW and a rated wind speed of $11\,\mathrm{ms}^{-1}$. In order to calculate the AEP, each research partner implemented their own script according to the 'Time series method' described in Section 2.1.4. The results are shown in Fig. 11 in terms of the percentage differences from the 'theoretical' value. All the results are significantly lower than the theoretical value. The differences in AEP are due to the differences in simulation results as well as the differences in the AEP methods between research partners. For WF-3 (CFX), a comparison was additionally made with and without the effects of turning (as described in Section 2.1.4). A very small difference due to the effects of flow turning was observed, shown by the diffence between WF-3 and WF-3T. Details of the variations of the AEP for each wind direction sector can be found in the Appendix as well as in Barber et al. (2021). It was found that a large variation between model and sector exists, without a particular trend.

## 4.2 Site 2

For this site, the AEP has been calculated at the validation location using the power curve of the Siemens SWT-DD-130 wind turbine. A hub height of 50 m instead of the real hub height of 115 m was used in order to avoid additional inaccuracies due to the wind speed profile vertical extrapolation. This wind turbine has a rated power of 3.9 MW and a rated wind speed of $13\,\mathrm{ms}^{-1}$. The results in Fig. 12 show that the results match very well for WF-1 (WindPro), WF-2 (WindSim), WF-4 (Fluent RANS) and WF-5a (Fluent SBES), with deviations less than 3%. The WF-6 (PALM) result is significantly lower, with a deviation of 32%. This is due to the under-predicted wind speed shown in Section 3.

Details of the variations of the AEP for each wind direction sector can be found in the Appendix as well as in Barber et al. (2021). It was found that the AEP was under-predicted in the three most frequent sectors (120°, 270° and 300°), leading to an overall under-prediction of AEP. For WF-6 (PALM), the 270° and 300° sectors are especially strongly under-predicted, explaining the large deviation compared to the theoretical AEP.

### 4.3 Site 3

For this site, the AEP has been calculated at the validation location using the power curve of the Enercon E-40 wind turbine. This wind turbine has a rated power of 0.6 MW and a rated wind speed of $12\,\mathrm{ms}^{-1}$. A hub height of 50 m was assumed. The results in Fig. 13 show that the results are all under-estimated, due to the under-estimations of wind speeds in the most frequent sectors. The under-estimation is quite consistent for all models (between 10% and 15%) except E-Wind, which under-estimates the AEP by 24%.

Details of the variations of the AEP for each wind direction sector can be found in the Appendix as well as in Barber et al. (2021). It was found that the AEP is significantly under-predicted in the most frequent sector (240°), leading to an overall under-prediction of AEP. There is no particular pattern in the variation of AEP between each workflow. The very small effect of flow turning on the WF-4 and WF-5a (Fluent) results can be seen for some sectors.

### 4.4 Site 4

For this site, the AEP has been calculated at the validation location using the power curve of the Enercon E-82 wind turbine. This wind turbine has a rated power of 2 MW and a rated wind speed of $12.5\,\mathrm{ms}^{-1}$. The hub height is 78 m. The results in Fig. 14 show that, as opposed to the Site 3, the results for Site 4 are all over-estimated, due to the over-estimations of wind speeds in the most frequent sectors. The over-estimation is largest for WF-4 (Fluent RANS), and the SBES simulation in WF-5a reduce this slightly. The effect of flow turning is minimal. The over-estimation is small for WF-1 (WindPro) and WF-2 (WindSim).

Details of the variations of the AEP for each wind direction sector can be found in the Appendix as well as in Barber et al. (2021). It was found that the large over-prediction in AEP by the WF-3 and WF-4 workflows mainly occurs because of the over-prediction in the 240° sector, because this is the most frequently occurring sector. Additionally, it was found that an under-prediction of WF-1 (WindPro) and WF-2 (WindSim) in the 210° sector is probably compensated for by the over-prediction in 240°, because both of these sectors occur for a similar proportion of time.

### 4.5 Site 5

For this site, the AEP has been calculated at the validation location using the power curve of the planned wind turbine type. The results shown in Fig. 15 match very well for WF-4T (Fluent RANS) and WF-5aT (Fluent SBES), with deviations less than 6%. The WF-1 (WindPro) and WF-2 (WindSim) results are significantly lower, with deviations of 30% and 26%, respectively. These deviations may initially be surprising, considering that the wind profile RMSE values between simulations and measurements

for the four workflows are similar. However, this has occurred because both under-predictions and over-predictions of the wind speed lead to a positive RMSE, whereas a combination of over-predicted and under-predicted wind speeds cancel each other out for the AEP calculation.

    Details of the variations of the AEP for each wind direction sector can be found in the Appendix as well as in Barber et al. (2021). It was found that shows that the AEP is particularly under-predicted in the 210° sector for all workflows, and in the

410 240° sector for WF-1 and WF-2, but not for WF-4T and WF-5aT.

Thus, the large difference in workflow results is attributed to the strongly under-predicted wind speed in the 240° sector of WindPro and WindSim. This has such a large influence on the AEP because of the very high average wind speeds in this sector combined with the high frequency of occurrence. It reveals an important point about this work – that the average absolute difference in wind speed over all sectors cannot be used as a reliable metric for the expected difference in AEP, even when it is weighted for the frequency of occurrence.

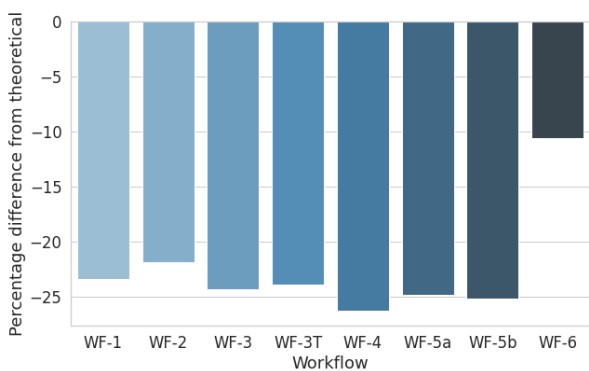

**Figure 11.** Site 1 - Percentage difference in gross AEP from the 'theoretical' value for each workflow applied

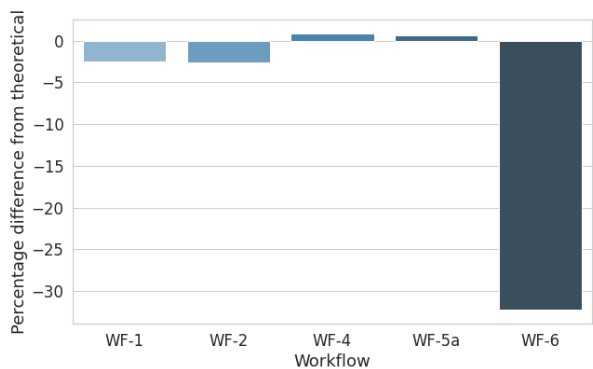

**Figure 12.** Site 2 - Percentage difference in gross AEP from the 'theoretical' value for each workflow applied

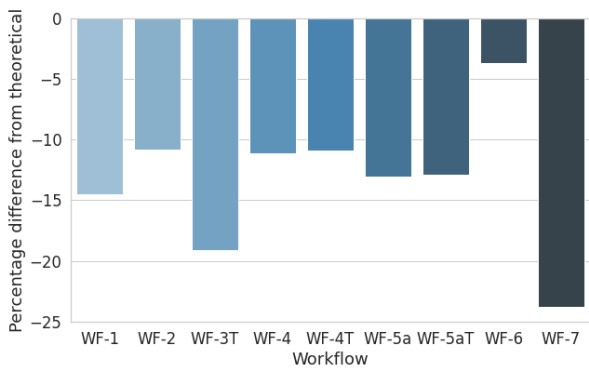

**Figure 13.** Site 3 - Percentage difference in gross AEP from the 'theoretical' value for each workflow applied

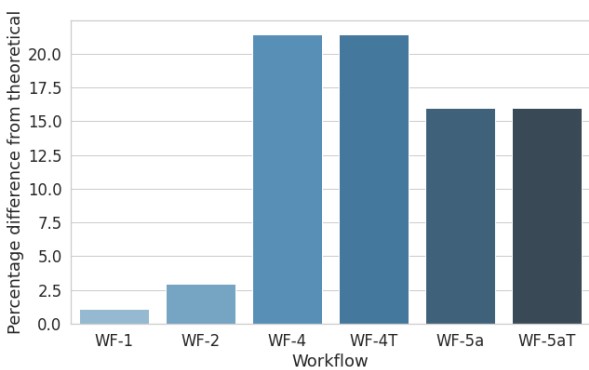

**Figure 14.** Site 4 - Percentage difference in gross AEP from the 'theoretical' value for each workflow applied

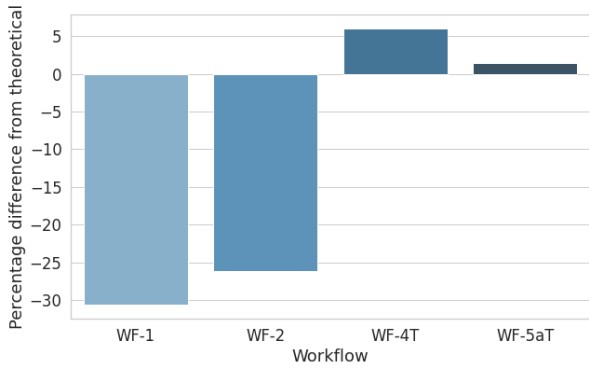

**Figure 15.** Site 5 - Percentage difference in gross AEP from the 'theoretical' value for each workflow applied

## 4.6 Comparison of wind speed and AEP results

In this section, the results of the simulated wind speeds and AEP calculations are compared by first plotting the error in simulated wind speeds against the error in expected AEP at the validation location for each site (Fig. 16). Each row corresponds to one site (from Site 1 to Site 5). The left-hand plots show the average error of all 12 wind speed direction sectors using a simple average, whereas the right-hand plots show the average error of all 12 wind speed direction sectors using an average weighted with wind speed frequency.

The correlation coefficients $R^2$ are summarised in Table 10 for both a simple average of the wind speeds and for a weighted average. It should be noted that the low number of points in these correlations, especially for Site 4 with only four points, could lead to the low correlation coefficients. The results should therefore only be used indicatively rather the quantitatively. Generally, the correlation between wind speed error and AEP error is fairly low quality, and $R^2$ ranging from 0.01 to 0.56 for Sites 1, 2, 3 and 5. The exception is Site 4, which has $R^2 = 0.94$ for simple averaging and 0.95 for weighted averaging. The authors could find no particular reason why Site 4 should result in this significantly better correlation, except for the seemingly coincidental combination of a wide range of different effects that have come together to produce this result. The fact that Site 4 involved wind speed data directly from the wind turbine, rather than at a met mast, cannot explain this, unless the wind speeds were altered by the operator or manufacturer without the knowledge of the authors. The weighting does not have a large effect on the correlation coefficients, and in some cases actually reduces the values. This indicates that the absolute value of the wind speed has a larger effect on the AEP accuracy than the actual accuracy in each sector, and can be explained by the cubic relationship between wind speed and power. Figure 17 shows the results from all sites on a single plots; on the left for simple averaging and on the right for weighted averaging. This increases the overall correlation coefficient to 0.19 and 0.15 for simple averaging and weighted averaging, respectively.

**Table 10.** Correlation coefficients and main set-up differences between wind speed and AEP errors for all sites

| Site | Site 1 | Site 2 | Site 3 | Site 4 | Site 5 |
|---|---|---|---|---|---|
| $R^2$ simple average | 0.04 | 0.35 | 0.01 | 0.94 | 0.56 |
| $R^2$ weighted average | 0.01 | 0.02 | 0.05 | 0.95 | 0.37 |
| Calibration data | Met mast | Met mast | Met mast | Wind turbine | Met mast |
| Validation data | Lidar | Met mast | Met mast | Wind turbine | Met mast |
| Distance cal. - val. | Near | Medium | Near | Near | Far |
| Measurement time | 1 year | 3 years | 1 year | 10 years | 6 months |
| Height difference extrapolated (m) | 7 | 28 | 0 | 0 | Unknown |

For four out of five sites, there is no clear relationship between wind speed error and overall AEP error. Wind profile prediction accuracy does not translate directly or linearly to AEP accuracy. There could be several reasons for this. It could

relate to the specific conditions at the site, to differences in workflow set-ups between the sites as well as to differences in workflow AEP calculation methods, as summarised below:

– Specific site conditions:

– The AEP depends strongly upon the relative strength and occurrence of the wind speed in the most commonly-occurring wind direction sectors. The wind speed errors occurring in wind direction sectors that have high average wind speeds transfer to much higher errors in AEP due to the the cubic dependency of power on wind speed. Furthermore, the wind speed errors occurring in wind direction sectors that have a higher frequency than other

wind directions contribute more to the overall AEP errors. In some cases, these two situations compound, and wind speed errors occurring in wind direction sectors that have high average wind speeds **and** a high frequency or occurrence dominate the AEP errors. Conversely, high wind speed errors occurring in wind speed sectors that have low average wind speeds and/or low frequencies of occurrence become less important for the overall AEP error.

– The site complexity is expected to affect the performance of the workflows in different ways for different wind

direction sectors. For example, if a steep slope occurs between the calibration and validation locations in a particular wind direction, the accuracy will be reduced in this sector for flow models that cannot capture flow separation correctly. On the other hand, if a forest is located between the calibration and validation locations in a particular wind direction, the accuracy will be reduced in this sector for flow models that cannot correctly predict the canopy effect. This topic is discussed further in a separate paper by the same author (Barber et al., 2022), in which a new

method for quantifying the complexity of a site is investigated.

– Differences in workflow set-ups between the sites:

– As mentioned in the introduction, this present study did not involve a systematic study of the effect of all calculation steps on the wind speed and AEP errors. This was due to the focus of the study on the relationship between cost and skill as can be seen in Barber et al. (2022). This means that it is not possible to quantify the effect of different

set-ups on the correlation between wind speed and AEP errors. However, the main differences in workflow set-up between sites includes: type and accuracy of calibration and validation measurement, relative location of calibration and validation measurements, measurement time period, long-term extrapolation method, steepness of wind turbine power curve, vertical extrapolation method and height difference between calibration and validation location. In further studies, these factors could be varied systematically for a range of sites. In Table 10, some of the main

differences between the AEP calculation set-up between sites are summarised. However, no obvious relationship between the correlation quality and these factors can be seen.

– Differences in workflow AEP calculation methods:

– As shown in Table 2 in Section 2.1.5, differences in AEP calculation methods carried out by different organisations cannot be entirely ruled out, and are expected to be larger between different workflows that have not been specifi-

cally compared and adjusted within a research project such as this one. A systematic study on this topic would be useful in order to quantify this effect.

- As well as this, different AEP calculation methods exist, and have been shown in Section 2.1.5 to have a significant effect on the AEP calculation. These differences can occur when different types and quality of data is available to different organisations. For example, the 'Time-series method" can only be applied when time series data is
actually available.

The combination of these different effects, and possibly others, lead to the results obtained in this work. Despite the fact that this work did not involve a systematic study of these effects, the importance of examining AEP errors as well as wind speed errors in any comparison study has been highlighted. As well as this, the complexity of the combined factors contributing to WRA errors has been demonstrated - even without including wake effects and other losses. the results show that the wind
model that produces the most accurate wind predictions for a certain wind direction over a certain time period does not always result in the most suitable model for the AEP estimation of a given complex terrain site. In fact, the large number of steps within the WRA process often lead to the choice of wind model being less important for the overall WRA accuracy than would suggest by only looking at wind speeds. Not only this, but additionally it is not immediately obvious which sites have a high correlation between wind speed and AEP errors and which ones do not. Future work will involve systematic studies of these
effects.

It is therefore concluded that it is vitally important for researchers to consider overall AEP - and all the steps towards calculating it - when evaluating simulations of flow over complex terrain. This agrees with similar recent qualitative findings from the CREYAP2021 study[13].

---

[13]https://windeurope.org/tech2021/creyap-2021/

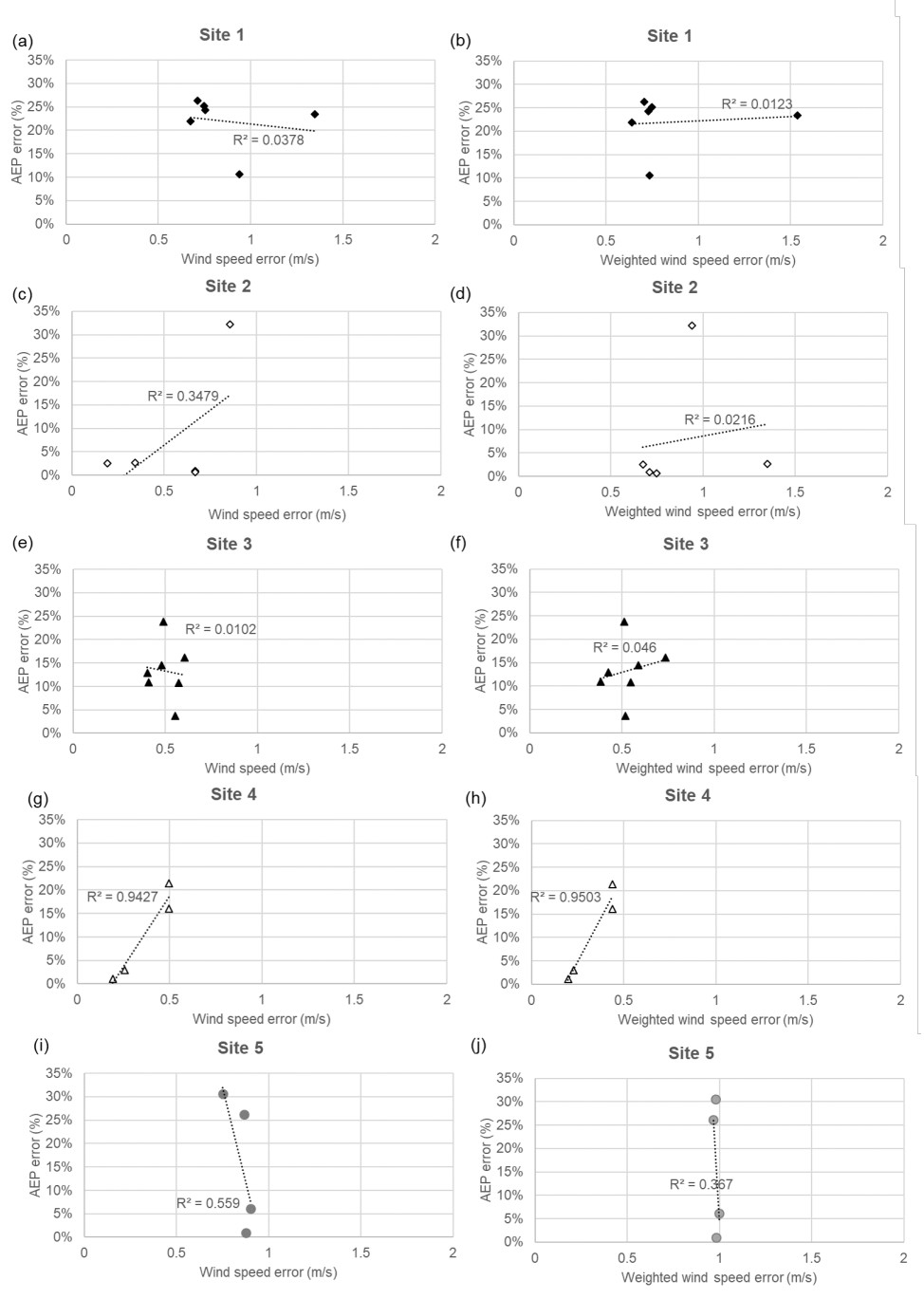

**Figure 16.** Comparison of AEP errors to wind speed errors for all workflows and all sites for each site separately (left: AEP compared to simple average wind speed; right: AEP compared to wind speed weighted for wind speed frequency

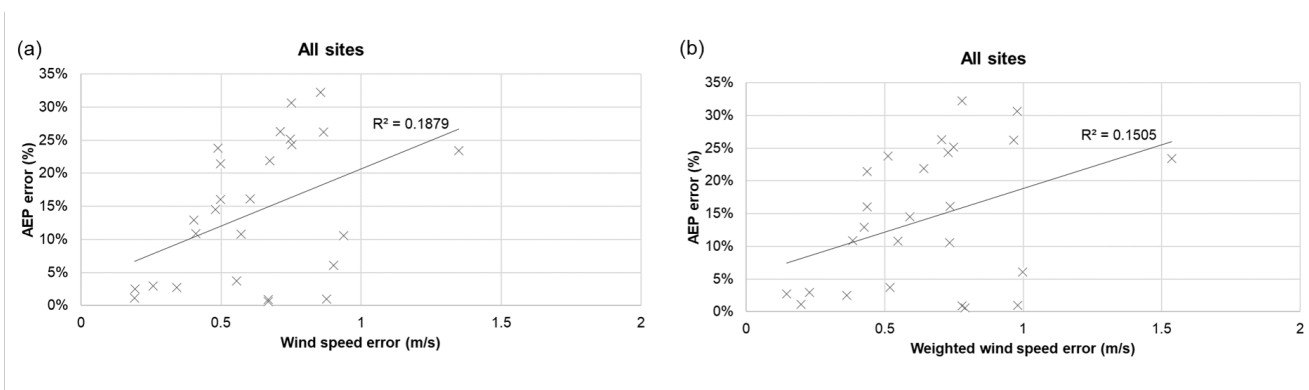

**Figure 17.** Comparison of AEP errors to wind speed errors for all workflows and all sites: (a) AEP compared to simple average wind speed; (b) AEP compared to wind speed weighted for wind speed frequency

## 5    Conclusions

A range of simulations have been carried out with seven different wind modelling tools at five different complex terrain sites and the results compared to wind speed measurements at validation locations. The study was then extended to AEP estimations (without wake effects), and it was found that wind profile prediction accuracy does not translate directly or linearly to AEP accuracy. This to the specific conditions at the site, to differences in workflow set-ups between the sites as well as to differences in workflow AEP calculation methods. Although a systematic study of the effect of all the possible varying factors was not

done, the work highlighted the importance of examining AEP errors as well as wind speed errors in any comparison study. As well as this, the complexity of the combined factors contributing to WRA errors has been demonstrated - even without including wake effects and other losses. The results show that the wind model that produces the most accurate wind predictions for a certain wind direction over a certain time period does not always result in the most suitable model for the AEP estimation of a given complex terrain site. In fact, the large number of steps within the WRA process often lead to the choice of wind

model being less important for the overall WRA accuracy than would suggest by only looking at wind speeds. Not only this, but additionally it is not immediately obvious which sites have a high correlation and which ones do not. Future work will involve systematic studies of these effects. It is therefore vitally important for researchers to consider overall AEP - and all the steps towards calculating it - when evaluating simulations of flow over complex terrain.

# 1 Appendix A1

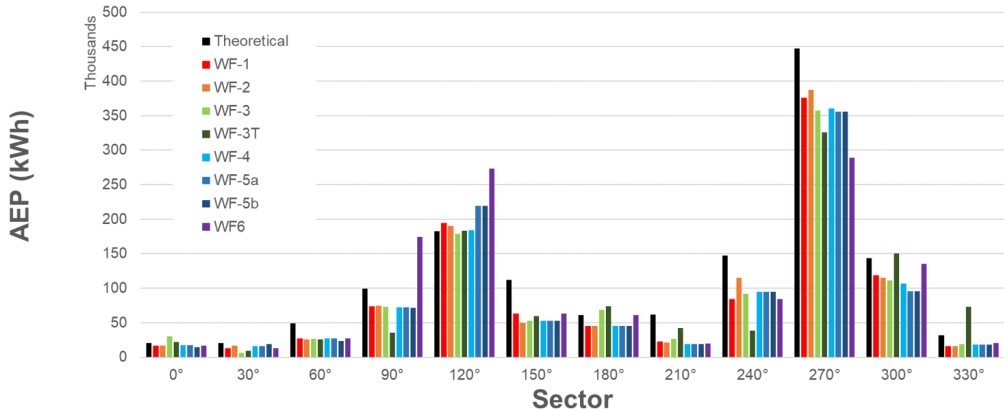

**Figure 1.** Site 1 - AEP per wind direction sector for each workflow applied

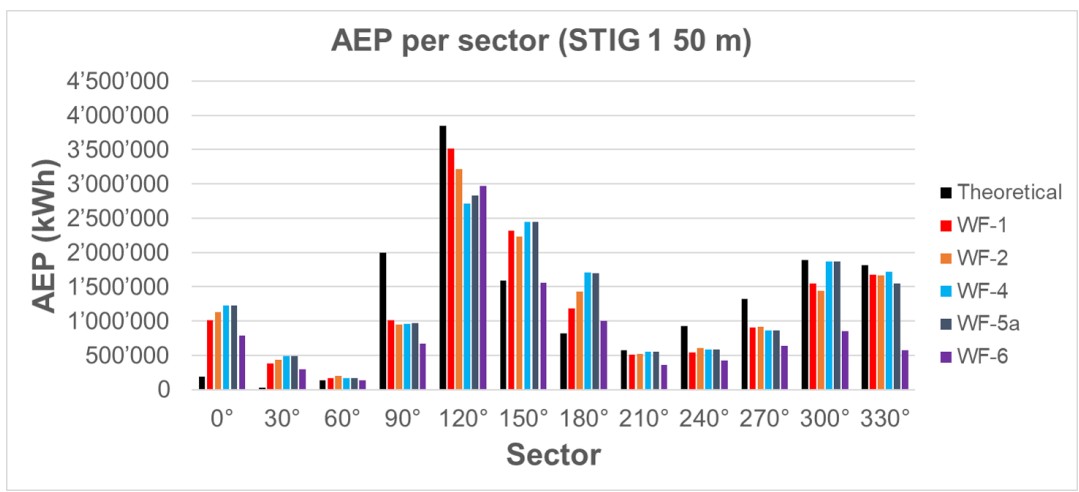

**Figure 2.** Site 2 - AEP per wind direction sector for each workflow applied

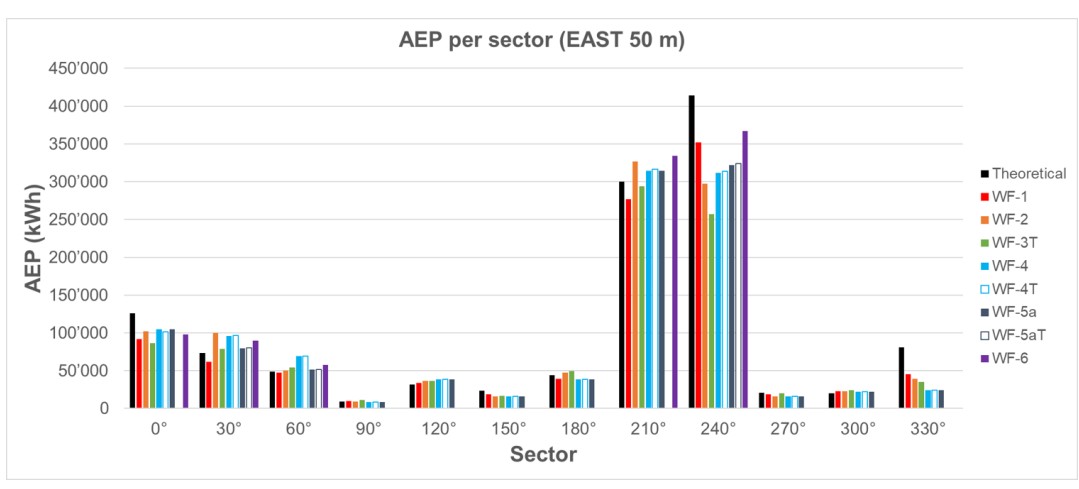

**Figure 3.** Site 3 - AEP per wind direction sector for each workflow applied

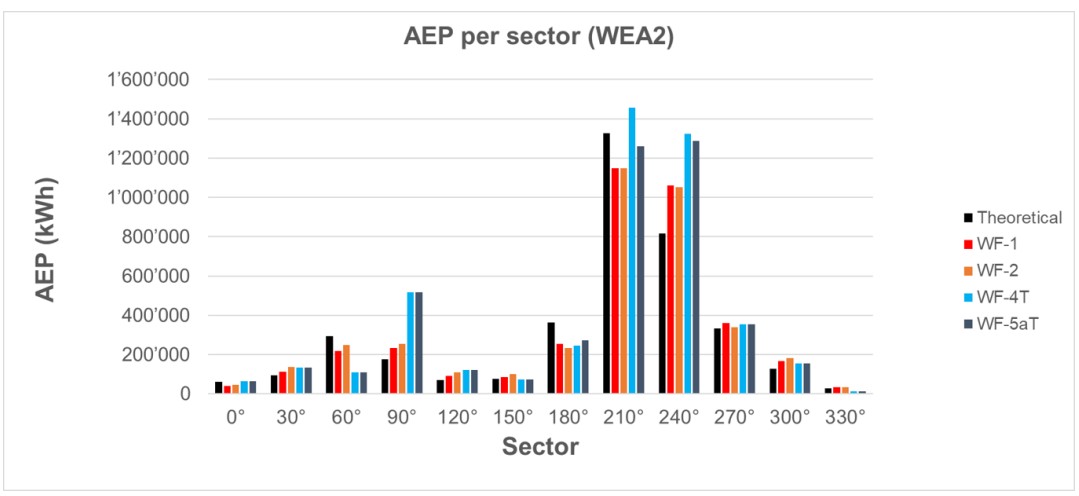

**Figure 4.** Site 4 - AEP per wind direction sector for each workflow applied

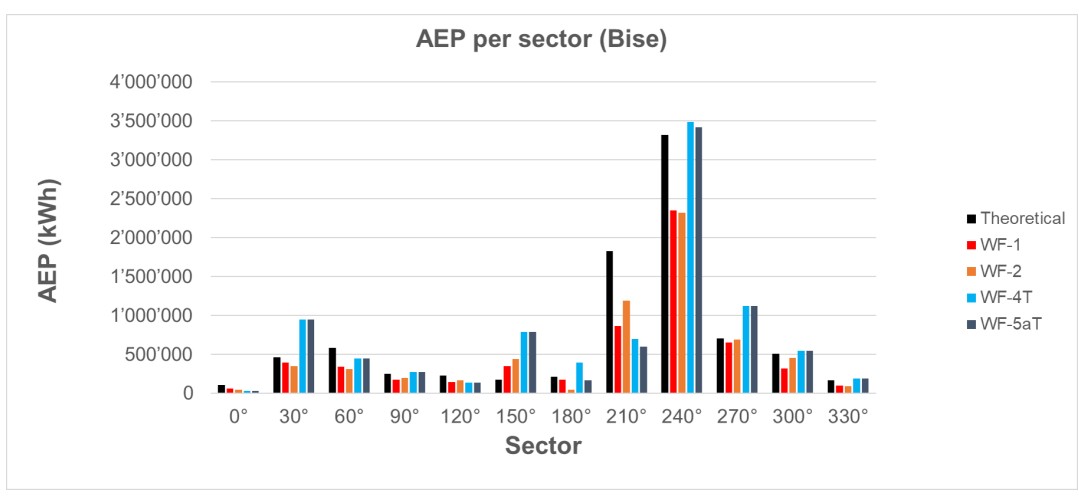

**Figure 5.** Site 5 - AEP per wind direction sector for each workflow applied

*Author contributions.*   Sarah Barber managed and coordinated the project and was responsible for writing the paper. Alain Schubiger carried out the Fluent simulations and developed the script for the AEP calculation process. Sara Koller and Dominik Eggli carried out the WindPro, WindSim and PALM simulations. Alexander Radi carried out the E-Wind simulations. Andreas Rumpf carried out the CFX simulations and developed the script for the AEP calculation process. Hermann Knaus supervised the work of Andreas Rumpf and provided valuable inputs for the paper.

*Competing interests.*   There are no competing interests

*Acknowledgements.*   This work was carried out as part of the projects "A new process for the pragmatic choice of wind models in complex terrain" funded by the Swiss Federal Office of Energy (SI/501807-01) and the German Federal Environmental Foundation (34933/01). Thanks also to the partner companies of this project for providing data and inputs about the simulated sites, including ewz, ADEV, Enercon as well as the Institut für Flugzeugbau at the Universität Stuttgart for providing us with the lidar data from the "Lidar complex" project.

on the left margin, 510 on the left margin

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
