# Peer review of "The wide range of factors contributing to Wind Resource Assessment accuracy in complex terrain"

_Wind Energy Science, 2021_

## Author Comment (AC1)

**wes-2021-158**

**Response to reviewer (Andrea Vignaroli)**

Thank you for taking the time to review this paper. We hope that we have answered your comments satisfactorily and look forward to further comments. In the marked-up version, the changes made relevant to your comments are marked in red. The blue changes have been made in response to the other reviewer.

**General comments**

Dear Barber et al.,first of all I'd like to  say that I completely agree with the underlying motivation for this work. It is important to focus on how inaccuracies in wind speed predictions translate in energy and show that it's a complex matter with a lot of variables (like wind direction, wind speed distribution, shear…) impacting the final results. It is very easy to be "right for the wrong reason". This is why so many studies focus on wind speed in one direction. The aim of those studies is to improve flow models and the only thing the scientific community can do is to design and conduct experiments while minimizing the amount of variables that could make the interpretation of results impossible.

**Specific comments**

As I said, the motivation and scientific question behind this work is very relevant. However I notice some decisions in your implementations which confuse me a bit. I don't understand if you decided to obtain results by including some of the uncertainty components of the AEP assessment process on purpose or you tried to avoid them. For example, you decided to base your results and comparison on long term corrected wind speed data being hopefully aware of  increased uncertainty that you take with you during the AEP comparisons. How do you know that certain differences in AEP are not due to  a long term correction far from perfect? Ok, but let's say that we want to include  long term correction uncertainty on purpose to see how it  translates in AEP, why did you decide to avoid vertical extrapolation from the measurement height to the hub height. Uncertainty of vertical extrapolation is another important source of uncertainty in the AEP assessment process. I would have avoided or included both.

- This is a good point, thank you for noticing this. It originates from the fact that the focus of the study was on the general comparison of costs and accuracy for the development of a WRA decision tool as described here: https://www.mdpi.com/1996-1073/15/3/1110 and that we were not entirely aware of the extent of the discrepancy between wind speed and AEP errors before carrying out the project. This means that the study wasn't designed specifically with the goal of investigating discrepancies between wind speed and AEP accuracies. If it had been, we would certainly have taken a systematic approach to all the possible sources of error.
- The important question is now what we do about this problem in this paper? Such a systematic study is certainly not possible at this stage (the project has finished and there is no funding yet for a future project). Our suggestion is to make it clear that this work does not involve a systematic study aimed at quantifying and understanding all possible sources of discrepancy between wind speed and AEP accuracies, but instead aims to (a) make readers aware that this problem exists and (b) highlight some of the different factors that could lead to this discrepancy. We have re-written some parts (Abstract, start of Section 2, Section 4) in order to achieve this. Please let us know what you think.

I think that using wind speeds measured by nacelle anemometers for the purpose of the article is quite a stretch. It would require quite a lot of analysis (flow inclination, rotor speed, pitch settings) in order to make the statement "but the ration will still be valid". I would have not used site 4.

- We do agree that site 4 is very different to the other sites for the reasons you mentioned. However, it would actually be interesting to understand the effect these differences may have on the wind speed and AEP error. Obviously there are far too many varying input conditions in order to do this within this work. It could be the topic of a further systematic study as discussed in the previous comment. If we now consider the new aim of the paper now to be "highlighting some of the factors that could lead to a discrepancy between wind speed and AEP accuracies" as discussed above, then site 4 could be included for the same reasons. We have adjusted the description of the results for this site accordingly (Section 4).

I am a bit puzzled how you can obtain a non zero error when you compare the wind speed at the calibration location when you consider one height only. (figure 2)

- Good point – we were puzzled about this too. The reason is actually because of interpolation errors. The location of the calibration measurement is not exactly correct in the simulations because it has to be located at a grid point (horizontally and vertically). This causes a small difference in wind speed at the calibration location. We have added some comments on this in Section 3.

My last specific comment is that you used given power curves for different wind turbine models for different sites. I assume that each of them are different with respect to generator/rotor area ratio and they will have different rated wind speed. Would it have been better to use only one for all sites so that the results are not affected by the power curve steepness? Given power curves are also tricky because they almost always need site specific adjustment. One way to make the study power curve independent would have been to use WPD (wind power density) as a metric instead of AEP.

- This is also a very good point that we did not think of for the same reasons as with your first comment. Now you mention it, we think that the steepness of the power curve will definitely have an effect on the results. We will definitely consider this if we get funding to do a more systematic study.
- For this paper, we again suggest resolving this issue by making it more clear that this work does not involve a systematic study aimed at quantifying and understanding all possible sources of discrepancy between wind speed and AEP accuracies, but instead aims to (a) make readers aware that this problem exists and (b) highlight some of the different factors that could lead to this discrepancy, as mentioned above. As well as that, we have now mentioned the topic in the analysis in Section 4.

**Technical corrections**

Line 51: It would be nice to mention that flow calculations in WindPro can be based on WAsP CFD (EllipSys3D) or WAsP linearized flow model (IBZ). I assume you used the IBZ model.

- Yes, we added this.

Line 56: WindSim can simulate more directions . But 12 were used for this analysis.

- OK, we changed this.

Line 107: I am missing some details of the MCP method used (linear least square, matrix, etc) and some metric for the reader to evaluate the accuracy of such a step (maybe a table with R^2,  measured and long term corrected mean wind speed?).

- We have created a new table for this information (Table 1).

Line 159 and Table 1: i don't think you explain the meaning of the abbreviation HSE or OST before using them

- You are right. We added this.

Line 260: Did you apply RIX correction? It's quite known that WAsP IBZ results need RIX correction for complex sites which will make a difference in terms of accuracy.

- No – we added this this to Section 1 and to the analysis in Section 4.

---

## Author Comment (AC2)

**wes-2021-158**

**Response to reviewer 2**

Thank you for taking the time to review this paper. We hope that we have answered your comments satisfactorily and look forward to further comments. In the marked-up version, the changes made relevant to your comments are marked in blue. The red changes have been made in response to the other reviewer.

**General Summary**

The authors conclude that in complex terrain, the wind energy community should consider the impact of wind direction to overall annual energy production in the wind resource assessment process. The authors display the results from 7 sets of wind flow models and 5 simulation locations. This sensitivity study leads to some interesting results, yet the authors should elaborate on their thought process and their findings. The authors also omit some key details to support their arguments and to uphold scientific reproducibility.

**Major Comments**

Section 2.1.1: It would be very useful to use a table to summarize the similarities and differences among the wind models. For example, both WF-1 (WindPro) and WF-2 (WindSim) calculate TI directly using the input met mast data. For a sample table, the columns can be WF-X's, a row can be "TI calculation", and the respective WF-1 and WF-2 elements can contain the same information. Such table can be constructed in many ways, at the discretion of the authors.

- This is a good idea, thanks. We have inserted a table as suggested, see Table 3.

Moreover, in a separate table, the authors should also discuss the pros and cons of different wind models, such as computational time (relating to line 84), labor required to set up a model run, recommended resolution, specific assumptions made in the code, when would an analyst use one model over the other, etc. This would guide the readers on how the wind models differ.

- This is also very important, and it actually relates to the overall goal of this project, as published here: https://www.mdpi.com/1996-1073/15/3/1110. We have put this in the same table as above and referred to the other paper for more details.

Section 2.2: Similarly, a table to summarize the 5 (or 4, since Site 5 is confidential) simulated sites would be useful. The table shall include heights of measurements, the duration of measurement period, the number of turbines, etc. For Site 2, it is confusing that its measurements seem to be available at multiple heights, and in Section 3.2 only the measurement of 1 height is discussed.

- We have created a new table as suggested (Table 9).
- It is explained on lines 226-227 that the data at different heights couldn't be used due to errors.

Lines 257 to 259: Can the authors explain why the validation errors are substantial? Given the model errors are very similar across the models (except for WF-1), what insights can we derive from this? Along the same line, Site 1 does not seem to be the outlier, because Sites 3 and 5 record validation errors of similar magnitude.

- For site 1, this discrepancy seems strange to us, as the calibration and validation locations are close. We suspect a problem with the lidar data, and have tried to discuss this with the data providers. However, we did not manage to find the problem

and do not want to publically suggest a problem with the data kindly provided to us by a partner. We have approached them but received no answer yet. We've therefore added a couple of comments and a contour plot to discuss this topic (line 297-300).
-   The validation errors for Sites 3 and 5 probably occur for different reasons, which we have explained now on lines 308-311, 321-325, 333-336 and 342-344, respectively.

The manuscript lacks consistency in analyzing the workflows and the sites. In lines 218 to 219, the authors reasoned why Site 4 does not need long-term extrapolation. In lines 314 to 315, the authors ignore the importance of vertical extrapolation in a project that focuses on complex terrain, which does not sound convincing. The authors should discuss under what circumstances a typical process is skipped for a site and explain why. Given the specific treatments each site requires, summarizing the information using a graphic or a table would be useful.

-   Thanks for noticing this, it does seem inconsistent. This comment relates strongly to the first point brought up by Reviewer 1. These inconsistencies originate from the fact that the focus of the study was on the general comparison of costs and accuracy for the development of a WRA decision tool as described here: https://www.mdpi.com/1996-1073/15/3/1110 and that we were not entirely aware of the extent of the discrepancy between wind speed and AEP errors before carrying out the project. This means that the study wasn't designed specifically with the goal of investigating discrepancies between wind speed and AEP accuracies. If it had been, we would certainly have taken a systematic approach to all the possible sources of error. Instead, the applied methods and steps were chosen according to the information available to us, the needs of each customer at each site, and the time available in the project. Some of the decisions we made appear inconsistent from the point of view of this paper.
-   We suggest dealing with this difficulty by making it clear that this work does not involve a systematic study aimed at quantifying and understanding all possible sources of discrepancy between wind speed and AEP accuracies, but instead aims to (a) make readers aware that this problem exists and (b) highlight some of the different factors that could lead to this discrepancy. In order to achieve this, we have re-written some parts (Abstract, start of Section 2).
-   As well as this, we have specifically mentioned the topic now in Section 4.

Wind direction is emphasized in the Abstract, but among the plots and tables throughout the manuscript, only the wind roses in Figure 1 mention wind direction. For example, lines 318 to 321 discuss AEP differences among wind direction sectors, without referring to any plots or tables to support the arguments. Similar problems can be seen from lines 328 to 331, from lines 338 to 342, from lines 351 to 357, and from lines 387 to 389.

-   In order to focus on the message of this paper and on request of the editor during the pre-review stage, we chose not to include further analysis of the wind profiles and of the wind directions that has been carried out and is presented in the 128-page project report. The discussion of AEP differences you mention all include references to plots inside this report (which is publicly available at the link provided).
-   However, based on your comment we decided that it would be better to include these particular plots in the appendix, because they are referred to in the text.

Moreover, the authors should emphasize the role of wind direction sectors in AEP calculation in more detail earlier, in which its role is not introduced until Section 2.1.5.

- We added a paragraph about the whole process, including the role of wind direction, in the introduction (lines 27-39).

The authors include and illustrate the (wind-direction) weighted wind speed results in some parts of the paper, but the current analysis does not fully support the arguments made in the text.

- We referred to the AEP vs. sectors plots in the final report because we didn't want to include too much information in the paper, as this would retract from the overall message (on request of the editor in the pre-review). However, we have now added these plots to the Appendix and hope this is OK.

Lines 369 to 379 contain the key message of the paper. The authors should also discuss which parts of the WRA process in their case studies that lead to the low correlation between wind speed error and AEP error. What should readers focus on among all the steps in the WRA process? Which steps of the WRA process are embedded with the most sensitivities?

- We have added a discussion on this in Section 4.

Each panel in Figure 12 consists of few data points, and the argument of low correlation between wind speed error and AEP error is partially a product of the lack of data samples. For instance, the authors fit a linear regression with only 4 data points in Figure 12 (g) through (j). Strictly speaking, such technique and visualization does not treat statistics properly. The authors should address the issue of low data samples in the text. One alternative is to examine the correlation between wind speed error and AEP error by combining the data across the 5 sites.

- We have done both these things: we mentioned this as an issue and then we plotted all the results together in a new Figure (Figure 17). As well as these, we have included a table summarizing the correlation coefficients (Table 10).

Overall, the manuscript needs a careful check on copyediting: line 137 uses "1 ms-1" and line 150 uses "1 m/s". The naming convention of the wind sites and model runs is also not uniform. For example, WF-5aT is used in lines 345 and 352. Is it equivalent to WF-5b, which is only found in line 86 throughout the manuscript? The reference style of the citation is sometimes incorrect, as seen in lines 26 and 170.

- Corrected. The usage of the "T" is now explained at the start of Section 4 (lines 354-356).

Minor Comments

Line 11 to 12: This sentence is confusing, please consider rephrasing it.

- Changed

Line 17: The brackets are not necessary.

- Removed

Line 23: The authors should also briefly explain what the steps, data types, and organizations are.

- Done

Line 27 to 28: Why does the full name of CREYAP use double quotes but 'complex' (line 20) and 'workflows' (line 46) use single quotes? Please be consistent.

- Removed

Line 34 to 35: Use "In their work" instead? "In this work" can be interpreted as the work done in your manuscript.

- Done

Section 2.1.1: What is the default or available number of wind direction sectors for WF-3 and WF-4?

- We developed a script for this. We added a sentence describing this.

Line 85 to 87: What are the differences between WF-5a and WF-5b?

- The difference is the number of sectors simulated in SBES. We have reworded this to make it more clear.

Lines 94 to 95: WF-7 can use more descriptions.

- Added, as well as a reference.

Lines 99 to 102: Consider splitting the sentence into two.

- Done

Line 99: Which of the WF-X's are counted as CFD simulations? Is WindSim considered as one?

- Yes, and it also says this in the WindSim description. We haven't changed anything here.

Line 128: What is "speed-up factor"? Is it simply the wind speed difference between the validation location and the calibration location?

- It's the ratio. We added a description on line xx.

Line 142: What is "10/60 hours"?

- This just refers to 10 minutes in units of hours, i.e. 10 minutes divided by 60 minutes. We have removed this now to avoid confusion (line xx).

Line 153: Who are the research partners?

- We have added this (marked in red because the other reviewer asked the same thing)

Line 158: Is the "less than 5% variation" in terms of power, energy/AEP, or wind speed? Do the authors mean "more than 5%"? This sentence somewhat contradicts with lines 161 to 162 of "5%".

- We think this is confusing because the % differences were not included in Table 1. These are included now. We also specified that we are referring to AEP.

Figure 1: The legends of the map are too small, and the dot colors are blended with the topography color scheme.

- We have increased the size of the figure and hope this is more clear. We haven't changed the colours because this is connected to significant effort (the people working on the project have left). The new plots showing the calibration and validation locations in more detail (e.g. Figure 3) hopefully make up for this.

Lines 214 to 216: This is a critical assumption and needs more attention. Did the authors look at the turbine availability or operation log to verify such assumption?

- We have reworded this and also brought up the issue in Section 4.

Line 221 and 227: How about WF-7?

- We have added this.

Table 2 to 6: The "Wind model" row is not necessary, as they are explained in Section 2.1.1.

- Removed

Table 4: Is there a reason why WF-7 is only applied for 1 case?

- Yes, because Enercon only wanted to apply their model to their site. Added to line xx.

Line 233: Do the authors mean 2021 instead of 2001? Also seen in lines 291 and 309.

- Yes, changed

Figure 2: The authors can consider using the same y-axis scale for plots (a) and (b).

- We made a decision not to do this for all the figures 2-6, and to be consistent in not doing that. This is because some of the bars then appear really small. We decided not to change this.

Line 248: How many measurement heights were used and what were the heights? Same for lines 250, 254, and 255.

- Added

Line 345: What is the difference between WF-4 and WF-4T? Why is WF-4T used here but not WF-4? Does WF-4T relate to Calculation 4 or 4.1 in Table 1?

- Explanation added on line xxx.

Figure 4: Can the authors explain why the calibration errors are so low for WF-1, WF-2, WF-3, and WF-6? Similar patterns are seen in Figures 5(a) and 6(a).

- Yes, because of the calibration method described in Section 2.1.2. We added a comment about this.

Line 297: Do the authors mean "time series at the validation location"?

- Yes, changed!

Lines 365 to 367: This sentence is vague. Please explain what the "wide range of different effects" are.

- We've changed this section significantly, this should be clear now.

Lines 369 to 370: This sentence is confusing. How did the authors conclude "absolute wind speed has a larger effect on AEP accuracy" based on "weighting based on wind speed frequency does not change the correlation between wind speed errors and AEP errors"? The authors need to explain their logic more.

- This has been explained better now.

Line 378: Correlation between what?

- Added.

Figure 12: The axes labels should be "AEP errors" and "Wind speed errors" to avoid confusion.

- Done